# LLF-Bench: Benchmark for Interactive Learning from Language Feedback

## Abstract

We propose a formal setup of Learning from Language Feedback (LLF) and a new benchmark, LLF-Bench (pronounced as "*elf-bench*"), to evaluate the ability of AI agents to interactively learn from natural language feedback and instructions. LLF is essential for people, largely because the rich information provided by language feedback can help a learner avoid much of trial and error and thereby speed up the learning process. AI agents, thanks to being powered by Large Language Models (LLMs), can potentially benefit from language feedback during learning like people do. However, existing benchmarks do not assess this crucial capability. They either use numeric reward feedback or require no learning at all (only planning or information retrieval). LLF-Bench, the benchmark we introduce, is designed to fill this omission. It is a diverse collection of decision-making tasks that includes user recommendation, poem writing, navigation, and robot control. LLF-Bench implements several randomization techniques to ensure that the agent actually needs to *learn* in order to complete these tasks. In addition, LLF-Bench allows configuring the kind of information conveyed by the feedback (e.g., performance assessment, explanations or suggestions), which facilitates studying how agents respond to different feedback types. Together, these features make LLF-Bench a unique research platform for developing and testing LLF agents.

## 1 Introduction

Natural language is an intuitive medium for a person to teach an AI agent, since that is how humans learn from and teach each other. Compared to rewards – the feedback modality typically used in the reinforcement learning (RL) paradigm (Sutton & Barto, 2018) – language feedback can provide rich signals about agent behaviors beyond a quantitative measure of instantaneous performance. For instance, language feedback can explain why the agent's previous suboptimal behaviors should be avoided, rather than just punishing the agent without giving justification. Language feedback can also provide direct suggestions on how the agent can improve its future behavior, similar to action feedback used in imitation learning (IL) (Ross et al., 2011; Spencer et al., 2021). However, providing action feedback to a robot as has traditionally been done in IL requires a teleoperation setup, which might not always be feasible. Language feedback, on the other hand, can be given verbally by an ordinary user (Liu et al., 2023a). In recommendation systems, incorporating user feedback has been studied under coactive learning (Shivaswamy & Joachims, 2015). Reinforcement learning from human feedback (RLHF, (Christiano et al., 2017)) and preference learning (Rafailov et al., 2024) incorporate ranking-based, not verbal feedback.

We capture the essence of using language as a feedback modality in a new learning paradigm – *Learning from Language Feedback* (LLF). In an LLF problem, an agent interacts with a task environment and receives language instructions and feedback. At the start of an episode, the agent is first given a natural language *instruction* that describes the objective of the task, the rules, and (optionally) side information that may help solve the problem. After executing an action in the environment, the agent receives teacher *feedback* in natural language, which can be used as a learning signal. LLF generalizes reinforcement learning (RL) from return maximization to general problem-solving. Like RL, LLF focuses on sequential decision problems. However, in contrast to RL, an LLF agent does *not* receive rewards in numeric form and is *not* necessarily tasked with maximizing returns.

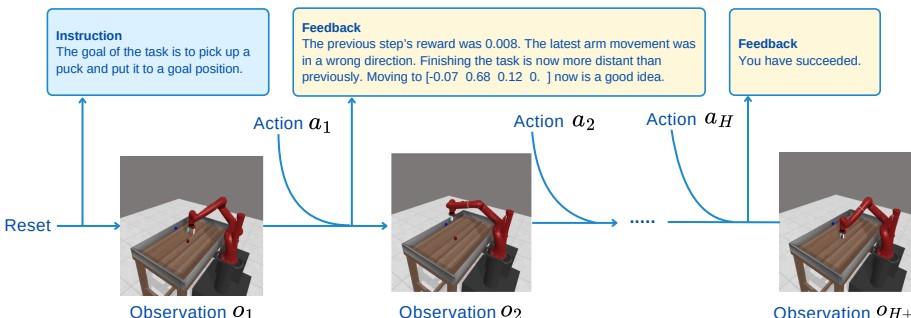

Figure 1: An example navigation task to illustrate our setup, Learning from Language Feedback (LLF). A single episode in LLF starts with a given instruction and can be multi-step long. The actions are taken by the agent that changes the observation and provides a *text feedback* to the agent. The agent receives no reward or any other form of feedback.

Figure 1 shows an example LLF flowchart. LLF replaces RL's assumption of numeric rewards with generic task instructions and feedback expressed in natural language. We can recover RL as an instance of LLF, e.g., with the instruction *"Maximize the accumulated rewards."* and the feedback template *"You've received a reward of X."*, under the assumption that the agent is prepared to parse the value of $X$ out of this template. But LLF covers many other scenarios that would be unnecessarily difficult to describe in the conventional RL framing, e.g., training a robotic arm controller by giving it general advice about the types of actions it should consider in certain situations, or asking an agent to write a poem in a certain style by showing a few examples.

In addition to the new learning paradigm, this work's contribution is LLF-Bench (Learning from Language Feedback Benchmark; pronounced as "*elf-bench*"), a simulation benchmark designed to evaluate an AI agent's ability to adapt quickly in LLF settings based on *just* language feedback. LLF-Bench is a collection of sequential decision-making problems, ranging from item recommendation to poem writing to robot control. Each of them has a natural-language description and a natural-language feedback generator that replaces RL's rewards as the learning signal (Section 3). Additionally, LLF-Bench provides a high-level wrapper that can convert any existing RL environment with OpenAI Gym interface into an LLF setup (Appendix C).

Prior to LLF-Bench, several benchmarks have been proposed to evaluate LLM-based agents for decision-making (e.g., AgentBench (Liu et al., 2023b), OpenAGI (Ge et al., 2023), MINT (Wang et al., 2023b), and LMRL Gym (Abdulhai et al., 2023)). However, most tasks in these benchmarks center around planning and information retrieval problems; only few require the agent to learn and adapt beyond what an LLM can already do. Also, many existing benchmarks lack language variations, so developers might accidentally identify a specific prompt that overfits to a particular verbal formulation of the task specification. This fails to reflect a key property of LLMs' real-life use cases, where a user needs LLM-based agents to handle tasks whose solution cannot be directly inferred from the task description and has to be learned from interactions and feedback instead (such as *"make the title text larger"* or *"wrap the code with an error-catching block."*). Are LLM-based agents capable of learning from general language feedback? LLF-Bench aims to provide a set of environments to help answer this question while addressing the challenges in reliable LLM agent benchmarking.

## 2 LLF: LEARNING FROM LANGUAGE FEEDBACK

We begin by introducing the Learning from Language Feedback (LLF) paradigm, and describe LLF-Bench in Section 3.

## 2.1 THE MECHANICS OF LLF

LLF is an abstract learning setup that models the interaction between an *agent* (e.g., a learning algorithm), a *world* (e.g., a robot hardware, or a recommendation system backed by a database), and a *teacher* (e.g., a person). The agent in the LLF setup is asked by the teacher to complete a task in the world via a natural-language *instruction*. The task's objective described in the instruction may be different from reward maximization and could include information about how to interpret observations, what the valid actions are, and what tips (such as examples) may help the agent solve the problem. After receiving the instruction, the agent sees the initial observation of the world state and starts to interact with the world by taking actions within the problem's prescribed action space (which can e.g. be a finite space, a continuous vector space, or a free-form text space just like that in RL). After an action is executed, the world's internal state may change and the agent sees the next observation of the world. As the agent interacts with the world, the teacher provides natural language *feedback* on how the agent performs to guide the agent to do better. This language feedback is a strict generalization of the reward signal in RL and can provide richer information to help agent learn (e.g., suggestions, explanations, etc.). If we group the world and the teacher in LLF together as an abstract *environment*, we see that LLF mainly replaces the reward maximization objective and numeric feedback in RL with a generic task instruction and language feedback. In LLF-Bench, we simulate LLF problems through the OpenAI Gym interface, described in Appendix C.

## 2.2 ISN'T RL ENOUGH?

The LLF setup is motivated by the inefficiency and unnaturalness of communicating intentions via rewards in the real world. The concept of return maximization in RL, while giving a simple abstraction of interactive learning, often creates a barrier for people to transfer knowledge and convey intentions to AI agents. The reward paradigm forces one to compress all the information one wishes to convey to the agent at a given step into a single numerical value expected to encourage or penalize certain behaviors. In addition, rewards are received only after the agent takes actions, so the agent has to not only learn to solve the task but *also* learn to understand the task's objective. This bottleneck limits the information that can be transferred to the agent and couples solution learning with intention understanding, causing the agent to learn inefficiently in a trial-and-error manner.

In many cases, it is also difficult for human designers to fully understand the long-term effects of maximizing return (the expected sum of rewards), even when each instantaneous reward makes sense. This misalignment has led to many surprising behaviors of RL agents (Amodei et al., 2016). Consequently, reward engineering has been a common practice in building RL systems, where the user iteratively tweaks the task's rewards by observing how the agent behaves after maximizing the current reward function. However, reward engineering is an expensive process. If agents were able to learn directly from language feedback, learning systems could be built more economically.

Overall, compared to RL, LLF embraces the rich language feedback used in human-to-human learning. Its expressivity provides a potentially more efficient mechanism for training agents than RL.

## 2.3 WHY SHOULD WE STUDY LLF NOW?

Interactive learning settings with language-based instructions (Misra et al., 2018; Chen et al., 2019) or observations have been extensively studied in the literature (Wang et al., 2016; Guu et al., 2017; Zhong et al., 2021). However, in all these settings, one assumes access to either gold actions or rewards. In contrast, in LLF the agent is provided with neither of these, which makes LLF appear harder than RL. We argue that this difficulty of working with general language feedback has been the reason why LLF hasn't received much attention previously, despite its potential benefits. Recently, Large Language Models (e.g., GPT4 (OpenAI, 2023), Gemini (Gemini Team, 2023)) have demonstrated impressive natural language processing abilities. In addition, multiple LLM agents have shown promising signs of solving text-based problems involving decision making, planning, information retrieval, and tool use (Wang et al., 2023a; Schick et al., 2023; Wu et al., 2023). Therefore, with LLMs, it may be possible to design algorithms to systematically solve general LLF problems. Conversely, solving LLF can also be viewed as a way to measure LLMs' ability to tackle new learning tasks.

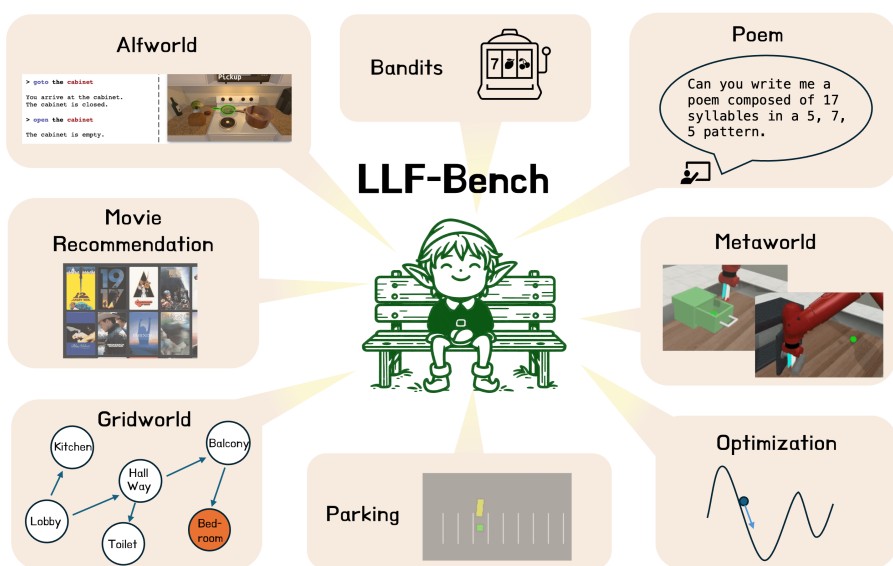

Figure 2: LLF-Bench ("Elf-bench") includes 8 sets of LLF problems. Image by Bing Chat.

In fact, with access to accurate LLMs, LLF is not harder than RL if the task instructions in LLF are detailed enough to allow the LLM to infer from observations alone (without language feedback) whether the agent has succeeded at following the instruction. (Note that this assumption does not mean that the instruction necessarily shows the agent how to solve the problem.) Under this assumption, LLF problems can always be solved without the feedback, by a reduction to an RL problem with sparse binary reward of success (the binary reward can be computed using a LLM to detect success based on the instruction and the observation). However, such a reduction approach would lead to inefficient learning.

## 3 LLF-BENCH

The main research question of LLF is how to best leverage the language feedback, which can convey more information than just success/failure, to learn the optimal policy for the task in a sample-efficient manner. We design LLF-Bench as a research platform to facilitate the development and evaluation of LLF agents (e.g., LLM agents) built to make progress on this research agenda.

### 3.1 PROBLEM SETS AND TASKS

LLF-Bench consists of 8 diverse sets of decision-making problems (see Figure 2), with different action spaces (discrete, continuous, and free-form text spaces) and decision horizons. Their brief descriptions follow below, with more details in Appendix B:

- `llf-bandit` is a verbalized version of the classic multi-armed bandit problem, which we implement based on gym-bandits. `llf-bandit` tests the agent's learning ability in an unknown environment with a finite number of actions.

- `llf-poem` consists of a set of poem writing tasks, where the agent needs to write a poem satisfying certain syllable- and line-constraints. These problems tests the agent's learning ability to infer and solve constraint satisfaction problems.

- `llf-reco-movie` simulates the scenario where a user wants movie or TV show recommendations based on some preferences. The user specifies their preferences in text, and any recommendation made by the agent is matched to a movie database for checking whether the preferences are matched correctly.

- `llf-optimization` consists of 8 loss functions (Rosenbrock, Bohachevsky, etc.) and provides an interface to give verbal feedback for the task of optimization on any loss function.

- `llf-parking` extends the Highway gym environment, providing a long-horizon goal-conditioned continuous control task. The agent must control an ego-vehicle to park in a given location without colliding with any obstacles in the environment.

- `llf-gridworld` evaluates the agent's ability to navigate in a graph-based environment. Each node of the graph is a room and the edges are doors connecting the rooms. The agent's goal is to navigate from the room it starts in to the room with treasure.

- `llf-alfworld` adds a wrapper on top of the Alfworld text-based environment (Shridhar et al., 2021) to provide language feedback instead of reward. In `llf-alfworld`, the agent is tasked to solve problems in a text-based house environment. The agent is tested for generalization as each episode can contain a new task in a new house environment.

- `llf-metaworld` is based on the existing Meta-World v2 benchmark (Yu et al., 2019) and supports both text (of low-dimensional states) and image observations. It comprises 50 simulated robotic manipulation tasks featuring a Sawyer arm and various objects that this arm needs to bring into desired configurations, such as opening doors, placing cubes in boxes, etc. As such, `llf-metaworld` is suitable for evaluating *Vision*-Language Models (VLMs) like GPT-4o, and we conduct studies of this kind in Section 5.

## 3.2 DESIGN OF LLF-BENCH

When designing a *learning* benchmark, an important consideration is whether the evaluation can truthfully reflect an agent's learning and generalization abilities and separate them from overfitting. To this end, we make two important design choices:

1. Following the framing of LLF, LLF-Bench implements the task instruction as part of the environment, as opposed to as part of the agent. The latter is common in the current literature of LLM agents, and many LLM agents heavily rely on using task-specific prompt templates (Yao et al., 2023; Wang et al., 2023a). Via this design, we encourage users of LLF-Bench to develop agents that can simultaneously work well across different problems sets in LLF-Bench. We hope that this paradigm shift will facilitate the development of more general learning agents that can solve multiple tasks, rather than agents tailored to a single task.

2. LLF-Bench provides the option to further randomize the textual description of task instruction and feedback that the agent receives. In addition, for several environments, we randomize the environment's latent parameters (e.g., to permute the action ordering in `llf-bandit` or change the room connectivity in `llf-gridworld`) when the environment is reset. Sensitivity to different phrasings of the same instruction is often used to measure the robustness of a text-based model (Ribeiro et al., 2018; Wallace et al., 2019). This design is motivated by the observation that LLMs *as of now* do not always perfectly understand semantics and can be sensitive to the exact texts that are presented (Zhu et al., 2023). It has been shown LLMs suffer from recency bias and can give drastically different outputs for semantically similar inputs (Arora et al., 2023; Leidinger et al., 2023). To combat that, for each problem instance in LLF-Bench, we manually curate a set of syntax templates via paraphrasing, which are used to produce a diverse yet semantically equivalent set of task instructions and feedback during interactions. Through randomization, LLF-Bench can better evaluate the agent's task solving ability and prevent the agent from overfitting a single text realization.

**Configurable Feedback System** One prominent feature of LLF-Bench is its configurable feedback system. Taking inspiration from the education research literature (Shute, 2008) and research on effective learning signals for reinforcement learning agents, such as heuristics-guided learning (Cheng et al., 2021) and hindsight learning (Sinclair et al., 2023), we classify the language feedback into 3 different types:

1. **Reward**: Feedback of performance on the current action (similar to reward scalars and success booleans, generalized and expressed via language). By using this feedback type, several classical RL environments can be comparably tested with LLF agents in LLF-Bench.

2. **Future Feedback**: Suggestions of future behaviors, such as hints (positive feedback) or things to avoid (negative feedback).

3. **Hindsight Feedback**: Explanation of past behaviors, such as why some behaviors are bad (negative feedback) or why some behaviors are helpful (positive feedback).

This taxonomy is inspired by the education research literature (Shute, 2008). While LLF-Bench can provide textualized numeric reward (i.e. the Reward type), learning from the Reward-type feedback is different from learning scalar rewards directly (e.g., in RL). Even when given textualized rewards, an LLM agent still has to understand the text and take actions accordingly, a challenge that is absent in RL where rewards are separately available as scalars. This is similar to the challenge of video-game agents that see the game score is on the screen but need a good semantic understanding of the screenshots to use it as a reward signal.

We also note that hindsight and future feedback types are different from text-based RL and language-grounding tasks. The latter two use only numeric feedback (if framed as RL) or actions (if framed as imitation learning). In LLF, the feedback is text. As Table 1 shows, text-based RL and language-grounding tasks are only similar to LLF in that they have observations that are text and/or images. However, in both cases feedback conveys a different type of information than observations: observations (partially) describe world state, while feedback says something about the agent's actions.

By default, an LLF-Bench environment provides a mix of these feedback types (when appropriate). It can also be easily configured to provide only a subset of these feedback categories. This makes for a more realistic learning problem, rather than the same type of atomic feedback at every step. LLF-Bench generates the feedback through templates. For each problem instance, we curated 5-20 versions of each atomic type of feedback. The environment, when queried, randomly samples from them and composes the samples together into overall feedback messages based on the configuration. Compared with generating feedback through an LLM-based simulated teacher, this template-based approach, while being less realistic, ensures reproducibility (through controlling the random seeds) and is efficient to run. See Appendix C.1 for details.

**Interface**    For ease of use, LLF-Bench adopts the OpenAI Gym API (Brockman et al., 2016), which abstracts the interaction with `reset` and `step` API functions. LLF-Bench environments return the natural language instruction and feedback as the observation (a Python dict) and the action spaces vary across problems. LLF-Bench environments also return rewards per the Gym `step` API. While agents in the LLF setup do not use rewards, the returned rewards can be used to evaluate an LLF agent's performance; this feature makes the LLF-Bench environments also usable as typical RL environments. LLF-Bench also provides a text-mode option (where both the observation and the action are free-form texts), so that it can also be used as a benchmark for evaluating LLMs as agents as well. Please see Appendix C for details of the API and the implementation of LLF-Bench.

## 4    RELATED WORK

In this section, we describe other benchmarks that focus on language-based agents. Works related to the LLF paradigm are covered in Appendix D.

**RL Benchmarks with Natural Language**    Many RL environments incorporate natural language. We provide a list summarizing their main features in Table 1. The RL environments can use language to describe the reward/goal (**instructions**), the **observations**, or the **actions**. Commonly, language is used as goal-specifying **instructions** (which is essentially a reward function) for an RL agent (e.g., GridLU by Bahdanau et al. (2019), ViZDoom Text by Chaplot et al. (2018), ISI Block by Misra et al. (2017), and Puddle World by Janner et al. (2018)). In this context, understanding and mapping instructions/goals to the state of the environment is the key challenge. Some RL environments naturally have **observations** in text; these include text-based adventure games (Text World by Côté et al. (2019) and NetHack by Küttler et al. (2020)) and HTML webpages (MiniWoB by Shi et al. (2017), MiniWOB++ by Liu et al. (2018), and WebShop by Yao et al. (2022)). Other RL environments have **action** spaces in text, i.e. an RL agent can generate a sequence of tokens as an action, such as a structured text representing a short executable program (e.g. SHRDLURN by Wang et al. (2016)). However, this was considered challenging due to the relatively large vocabulary space and the difficulty of learning to generate a sequence. None of these environments provide rewards as text

---

[1]The scalar reward is for evaluation, not for agent learning in the LLF setup.

| Environment | Observation Space | Action Space | Reward Space | Language Variations | Language Feedback |
|---|---|---|---|---|---|
| | Language Grounding Envs | | | | |
| SHRDLURN (Wang et al., 2016) | Vector | Text | Scalar | None | No |
| GridLU (Bahdanau et al., 2019) | Image | Discrete | Scalar | None | No |
| VizDoom Text (Chaplot et al., 2018) | Image | Discrete | Scalar | None | No |
| ISI Block (Misra et al., 2017) | Image | Discrete | Scalar | None | No |
| Puddle World (Janner et al., 2018) | Image | Discrete | Scalar | None | No |
| | Text-based Games | | | | |
| BabyAI (Chevalier-Boisvert et al., 2019) | Image | Discrete | Scalar | None | No |
| Zork (Narasimhan et al., 2015) | Text | Text | Scalar | None | No |
| TextWorld (Côté et al., 2019) | Text | Text | Scalar | None | No |
| NetHack (Küttler et al., 2020) | Image | Discrete | Scalar | None | No |
| | Web-Navigation Envs | | | | |
| MiniWoB (Shi et al., 2017) | Text/Image | Disc/Cont | Scalar | None | No |
| MiniWOB++ (Liu et al., 2018) | Text/Image | Disc/Cont | Scalar | Observation | No |
| WebShop (Yao et al., 2022) | Text/Image | Text | Scalar | None | No |
| | LLM Agent Benchmark Envs | | | | |
| AgentBench (Liu et al., 2023b) | Text | Text | Scalar | None | No |
| OpenAGI (Ge et al., 2023) | Text | Text | Scalar | None | No |
| MINT (Wang et al., 2023b) | Text | Text | Scalar | None | Yes (LLM) |
| LMRL Gym (Abdulhai et al., 2023) | Text | Text | Scalar | None | No |
| DialOp (Lin et al., 2023) | Text | Text | Scalar+Text | None | Yes (LLM) |
| MLAgentBench (Huang et al., 2023) | Text | Text | Scalar | None | No |
| **LLF-Bench** (Ours) | Text/Image | All | Scalar[1]+Text | All | Yes (Synthetic) |

Table 1: Comparison of decision-making environments that use natural language to instruct model behavior, represent observation, or is part of the action output. "Language Variations" refers to whether there are multiple descriptions of the same instruction, observation, or reward. "Disc/Cont" means the output is a mix of discrete and continuous variables. **LLF-Bench** offers text representation for instruction, observation, and reward, generates paraphrasing to prevent prompt hacking, and offers procedurally generated synthetic feedback for fast and cheap evaluation.

and do not provide feedback on actions. They also do not consider variations in language expressions – such as different phrasing or writing that represent the same underlying goal or state of the environment. Many of these environments are unsuitable for testing LLM agents due to having an observation space that is pixel or vector-based, and the types of tasks are dissimilar to what people use LLMs for today.

**LLM Agent Benchmarks**    Building agents based on LLMs has ushered in a new set of challenges. In general, the environments included in these benchmarks only require planning and information retrieval, and have sparse reward signals at the end of each attempt to solve the task. Very few of these benchmarks measure the ability of an agent to learn and adapt to a task (e.g., the Abstraction and Reasoning Corpus by Chollet (2019)). Liu et al. (2023b) proposed a set of environments that cover a few popular types of task setups, such as web browsing, game, and code generation. Their focus is on the diversity of tasks, not LLMs' robustness or ability to incorporate feedback – two factors crucial for LLMs' successful operation in user-centric environments. Ge et al. (2023) constructed a set of tasks where LLMs are prompted to use language or vision-related models to solve a complex task that requires multiple steps. The task-level feedback they provide is a numerical score from a domain-specific evaluation method. MINT (Wang et al., 2023b) is a benchmark that also offers natural language style feedback. However, MINT synthesizes user feedback by prompting LLMs. This incurs additional costs, introduces additional variability in the evaluation process, and makes it challenging to represent the diversity of human feedback styles. LMRL Gym (Abdulhai et al., 2023) provides a set of 8 environments that include full and partial observability. The tasks are similar to language-grounding tasks and text games. However, no interim feedback is provided during multi-round interactions. DialOp (Lin et al., 2023) provided three constraint-satisfaction-style planning tasks where an agent carries out a conversation with a human user. They collected a dataset with real human responses and noted LLM-provided responses have low quality and halluci-

| Problem sets and problems | gpt-4o | gpt-4-0125 -preview | gpt-3.5-turbo | llama3-70b | gemini-pro | phi3-mini-128k |
|---|---|---|---|---|---|---|
| **Bandits** | | | | | | |
| 10ArmedUniform | 1.38 (0.04) | 1.49 (0.08) | 1.35 (0.05) | 1.34 (0.04) | 1.40 (0.03) | **1.65** (0.10) |
| 10ArmedGaussian | 1.20 (0.27) | **2.21** (0.60) | 1.60 (0.24) | 1.19 (0.22) | 1.42 (0.24) | 1.46 (0.24) |
| **Optimization** | | | | | | |
| Booth | **-3.93** (0.68) | -100.46 (27.04) | -112.60 (9.55) | -38.25 (12.72) | -119.02 (17.12) | -493.96 (15.06) |
| McCormick | **-0.19** (0.05) | -0.40 (0.08) | -2.07 (0.28) | -0.72 (0.09) | -1.63 (0.20) | -2.49 (0.33) |
| Rosenbrock | -1.19 (0.33) | **-0.64** (0.16) | -344.43 (65.87) | -82.29 (35.32) | -306.49 (69.31) | -601.08 (60.37) |
| SixHumpCamel | **-0.23** (0.06) | -0.29 (0.22) | -5.60 (0.51) | -0.99 (0.28) | -3.15 (0.44) | -11.13 (0.38) |
| **Movie Rec.** | | | | | | |
| reco-movie | -5.28 (1.55) | -9.10 (3.48) | -7.17 (1.54) | -6.10 (1.49) | **-3.45** (1.13) | -14.23 (1.91) |
| **Highway** | | | | | | |
| parking | -14.32 (0.39) | -13.69 (1.04) | -14.53 (0.47) | -14.49 (0.51) | **-7.03** (0.97) | -13.94 (0.35) |
| **Poem** | | | | | | |
| Haiku | -6.59 (1.51) | **-0.80** (0.34) | -18.00 (1.87) | -14.94 (1.49) | -1.08 (0.29) | -4.58 (1.04) |
| Tanka | -9.03 (1.49) | **-0.36** (0.17) | -18.71 (1.83) | -24.04 (1.34) | -1.52 (0.33) | -11.56 (1.47) |
| LineSylConstr | -13.92 (1.33) | -23.49 (2.60) | -27.44 (0.57) | -25.01 (0.98) | **-0.37** (0.13) | -28.06 (0.43) |
| **Navigation** | | | | | | |
| gridworld | **1.00** (0.00) | **1.00** (0.00) | 0.70 (0.06) | **1.00** (0.00) | 0.92 (0.04) | 0.12 (0.05) |
| alfworld | 0.80 (0.06) | **0.86** (0.05) | 0.44 (0.07) | 0.78 (0.06) | 0.52 (0.07) | 0.00 (0.00) |
| **Meta-World** | | | | | | |
| reach | **1.00** (0.00) | **1.00** (0.00) | **1.00** (0.00) | 0.92 (0.04) | 0.54 (0.07) | 0.36 (0.07) |
| button-press-wall | 0.82 (0.05) | **0.90** (0.09) | 0.76 (0.06) | 0.88 (0.05) | 0.36 (0.07) | 0.02 (0.02) |
| bin-picking | 0.88 (0.05) | **1.00** (0.00) | 0.30 (0.06) | 0.52 (0.07) | 0.02 (0.02) | 0.00 (0.00) |
| pick-place | 0.68 (0.07) | **0.70** (0.15) | 0.30 (0.06) | **0.70** (0.06) | 0.10 (0.04) | 0.00 (0.00) |
| assembly | 0.00 (0.00) | **0.10** (0.09) | 0.06 (0.03) | 0.00 (0.00) | **0.10** (0.04) | 0.00 (0.00) |
| push | 0.86 (0.05) | 0.80 (0.13) | 0.56 (0.07) | **0.88** (0.05) | 0.00 (0.00) | 0.02 (0.02) |
| box-close | **0.88** (0.05) | 0.70 (0.15) | 0.36 (0.07) | 0.60 (0.07) | 0.00 (0.00) | 0.02 (0.02) |
| hand-insert | 0.16 (0.05) | **0.30** (0.15) | 0.28 (0.06) | 0.20 (0.06) | 0.02 (0.01) | 0.00 (0.00) |
| faucet-open | **1.00** (0.00) | **1.00** (0.00) | 0.84 (0.05) | 0.94 (0.03) | 0.26 (0.06) | 0.02 (0.02) |
| dial-turn | **1.00** (0.00) | **1.00** (0.00) | 0.92 (0.04) | 0.96 (0.03) | 0.48 (0.07) | 0.06 (0.03) |

Table 2: Mean and standard error of the return of the Basic Agent **with all feedback types** available to the agent. For GPT-4-0125-preview, because of cost, the statistics are computed over 10 episodes (except for Alfworld, for which, due to high problem instance variability, we used 50 episodes). For other language models, 50 episodes are used. For Meta-World, Alfworld, and Gridworld, the mean return is defined as the policy's success rate, which uniquely determines the standard error. Therefore, for the problems from these three problem sets, the st.e. is shown in gray.

nate. MLAgentBench (Huang et al., 2023) evaluates the ability of agents to build machine learning models, but no verbal feedback is provided.

## 5 EXPERIMENTAL RESULTS

To demonstrate the usability of LLF-Bench and the difficulty spectrum of its tasks, we experimented with state-of-the-art (SoTA) LLMs (GPTs[2] (OpenAI, 2023), Gemini (Gemini Team, 2023), Llama-3 (Touvron et al., 2023), Phi-3 (Abdin et al., 2024)).

**Agent and Setup** We use the TEXTWRAPPER provided with LLF-Bench to format observations and feedback into text, suitable for evaluating LLMs as agents. Then we implemented a Reflexion-based[3] basic agent (Shinn et al., 2023) that formats up to 20 most recent observation-feedback pairs into an LLM's context along with a system prompt as listed Figure 4 in Appendix. We conduct all experiments using API access to SoTA LLMs queried during the month of May 2024. All environments are initialized with horizon of $H = 30$ (i.e., the RESET function of the environment is called after 30 time-steps to initiate a new episode), and statistics are computed by 50 independent episodes. All experiments are run with the basic instruction (see Appendix C).

**Results** Table 2 shows the results of learning with full feedback of *all* types, and Table 3 shows the results of learning from a *restricted feedback set (Reward and Hindsight Feedback)*, which shares

---

[2]We use gpt-4o-2024-05-13 and gpt-4-0125-preview.

[3]Our implementation differs from the original Reflexion in that the original Reflexion implementation additionally stores the reflections in the agent's memory buffer but here we store the past observation-feedback pairs only. We found that this simplified version performs better. See Appendix E.1.

| *Problem sets* and problems | gpt-4o | gpt-4-0125 -preview | gpt-3.5-turbo | llama3-70b | gemini-pro | phi3-mini-128k |
|---|---|---|---|---|---|---|
| **Bandits** | | | | | | |
| 10ArmedUniform | 2.65 (0.13) | 3.18 (0.37) | 4.19 (0.59) | **4.44** (0.43) | 0.58 (0.05) | 1.65 (0.10) |
| 10ArmedGaussian | 1.17 (0.37) | 1.54 (0.76) | 1.62 (0.76) | 1.25 (0.53) | **2.03** (1.03) | -4.29 (2.42) |
| **Optimization** | | | | | | |
| Booth | -14.26 (1.81) | **-11.53** (3.27) | -125.93 (12.07) | -38.32 (6.64) | -112.57 (14.24) | -515.72 (15.58) |
| McCormick | **-0.26** (0.03) | -0.32 (0.08) | -1.91 (0.32) | -1.06 (0.16) | -1.38 (0.15) | -0.97 (0.18) |
| Rosenbrock | -10.51 (3.55) | **-1.49** (0.93) | -281.24 (64.88) | -73.01 (34.56) | -12.83 (6.22) | -317.71 (33.96) |
| SixHumpCamel | -0.56 (0.07) | -0.27 (0.14) | -6.03 (0.49) | -2.02 (0.48) | **-0.15** (0.04) | -9.86 (0.56) |
| **Movie Rec.** | | | | | | |
| reco-movie | **-6.98** (1.08) | -9.75 (3.59) | -11.88 (1.81) | -9.61 (1.73) | -18.74 (1.77) | -18.36 (1.97) |
| **Highway** | | | | | | |
| parking | -14.32 (0.28) | -13.24 (1.06) | -14.95 (0.45) | -14.49 (0.51) | **-7.87** (0.94) | -13.94 (0.35) |
| **Poem** | | | | | | |
| Haiku | -3.98 (0.82) | -7.63 (3.62) | -18.99 (1.76) | -9.02 (1.28) | **-0.67** (0.08) | -9.49 (1.48) |
| Tanka | -20.68 (1.35) | **-0.34** (0.19) | -20.77 (1.79) | -15.96 (1.55) | -1.34 (0.22) | -15.26 (0.94) |
| LineSylConstr | -23.97 (0.64) | -28.73 (0.36) | -28.44 (0.44) | -24.81 (0.98) | **-0.51** (0.17) | -28.45 (0.41) |
| **Navigation** | | | | | | |
| gridworld | **0.95** (0.02) | 0.70 (0.15) | 0.30 (0.06) | 0.92 (0.04) | 0.00 (0.00) | 0.10 (0.04) |
| alfworld | 0.64 (0.07) | 0.54 (0.07) | 0.18 (0.05) | **0.78** (0.06) | 0.30 (0.06) | 0.00 (0.00) |
| **Meta-World** | | | | | | |
| reach | **0.82** (0.04) | 0.70 (0.15) | 0.08 (0.04) | 0.16 (0.05) | 0.00 (0.00) | 0.02 (0.02) |
| button-press-wall | **0.56** (0.05) | 0.50 (0.16) | 0.00 (0.00) | 0.34 (0.07) | 0.00 (0.00) | 0.08 (0.04) |
| bin-picking | **0.00** (0.00) | **0.00** (0.00) | **0.00** (0.00) | **0.00** (0.00) | **0.00** (0.00) | **0.00** (0.00) |
| pick-place | **0.46** (0.05) | 0.20 (0.13) | 0.00 (0.00) | 0.02 (0.02) | 0.00 (0.00) | 0.00 (0.00) |
| assembly | **0.00** (0.00) | **0.00** (0.00) | **0.00** (0.00) | **0.00** (0.00) | **0.00** (0.00) | **0.00** (0.00) |
| push | 0.35 (0.05) | **0.90** (0.09) | 0.02 (0.02) | 0.06 (0.03) | 0.00 (0.00) | 0.00 (0.00) |
| box-close | **0.00** (0.00) | **0.00** (0.00) | **0.00** (0.00) | **0.00** (0.00) | **0.00** (0.00) | **0.00** (0.00) |
| hand-insert | 0.13 (0.03) | **0.20** (0.13) | 0.02 (0.02) | 0.04 (0.03) | 0.00 (0.00) | 0.00 (0.00) |
| faucet-open | 0.57 (0.05) | 0.40 (0.15) | 0.00 (0.00) | **0.94** (0.03) | 0.06 (0.03) | 0.10 (0.04) |
| dial-turn | 0.05 (0.02) | 0.10 (0.09) | 0.02 (0.02) | **0.84** (0.05) | 0.02 (0.02) | 0.00 (0.00) |

Table 3: Mean and standard error of the return of the Basic Agent **with Reward and Hindsight feedback types only**. For GPT-4-0125-preview, because of cost, the statistics are computed over 10 episodes (except for Alfworld, for which, due to high problem instance variability, we used 50 episodes). For other language models, 50 episodes are used. For Meta-World, Alfworld, and Gridworld, the mean return is defined as the policy's success rate, which uniquely determines the standard error. Therefore, for the problems from these three problem sets, the st.e. is shown in gray.

similarities with text-based RL environments. Table 2 establishes Basic Agent's performance when the feedback contains (nearly) all information required to act optimally, because the Future feedback explicitly tells the agent the (near-) optimal action to take, and the agent just needs to be "smart" enough to recognize this information among other, less useful feedback. On the other hand, Table 3 shows the agent's performance under the more difficult conditions, when the agent gets only indirect feedback. Thus, for a given LLM, we should expect its corresponding performance in Table 2 to be generally higher than in Table 3.

We observe that different environments test the capabilities of different LLMs (Table 2). For instance, GPT-4 variants perform the best in numerical optimization, whereas Gemini-Pro performs the best in temporally extended control problems like Highway parking. There is a definite benefit from model size. E.g., Phi-3-mini and GPT-3.5-turbo perform significantly worse than frontier models like GPT-4 or Gemini-pro across all tasks. However we observe that Llama3-70b can be competitive in Navigation and Bandit optimization tasks at a fraction of the cost of frontier models. Moving from Table 2 to Table 3 , we observe that the information in the feedback can significantly affect the learning quality of LLM agents. For instance, across all the Meta-World tasks, we observe a sharp decline in agent performance without the Future Feedback from the environment. However, on easier environments such as Bandits (black-box) and Poem (text editing), the best LLM performance is comparable across the different feedback sets, suggesting that the headroom to improve using Future Feedback is smaller in those environments.

In Table 4, we also report the experiments results of gpt-4o using image observations (in addition to text) in the Meta-World tasks. When all feedback types are provided, using image observation does not lead to better performance; but when Future Feedback (which suggests expert moves) is removed, using image observation improves the agent's performance. We note that the focus of our paper is not on evaluating whether images are useful for some tasks or designing the best vision-based agent, but instead on designing LLF-Bench to make studying such questions convenient.

| Models | gpt-4o (T+V) | gpt-4o (T) | Models | gpt-4o (T+V) | gpt-4o (T) |
|---|---|---|---|---|---|
| reach | **1.00** (0.00) | **1.00** (0.00) | reach | 0.64 (0.07) | **0.82** (0.04) |
| button-press-wall | **0.82** (0.05) | **0.82** (0.05) | button-press-wall | 0.36 (0.07) | **0.56** (0.05) |
| bin-picking | 0.72 (0.06) | **0.88** (0.05) | bin-picking | **0.00** (0.00) | **0.00** (0.00) |
| pick-place | 0.50 (0.07) | **0.68** (0.07) | pick-place | 0.26 (0.06) | **0.46** (0.05) |
| assembly | **0.00** (0.00) | **0.00** (0.00) | assembly | **0.00** (0.00) | **0.00** (0.00) |
| push | **0.88** (0.05) | 0.86 (0.05) | push | **0.52** (0.07) | 0.35 (0.05) |
| box-close | 0.74 (0.06) | **0.88** (0.05) | box-close | **0.00** (0.00) | **0.00** (0.00) |
| hand-insert | **0.18** (0.05) | 0.16 (0.05) | hand-insert | **0.20** (0.05) | 0.13 (0.03) |
| faucet-open | **1.00** (0.00) | **1.00** (0.00) | faucet-open | **0.78** (0.06) | 0.57 (0.05) |
| dial-turn | **1.00** (0.00) | **1.00** (0.00) | dial-turn | **0.16** (0.05) | 0.05 (0.02) |

(a) All feedback types          (b) Reward and Hindsight feedback types

Table 4: Mean and standard error of the success of the Basic Agent solving `llf-metaworld` tasks, which provide simulated camera images along with low-level states as observations, as shown in the *mean (st.e.)* format and computed with 50 episodes. (T) denotes **using only text observations**; (T+V) denotes **using both text and image observations**.

## 6 CONCLUSION

We introduced LLF-Bench to evaluate AI agents' ability to learn interactively from instructions and language feedback. We conjecture that the LLF paradigm will be significant for speeding up the agents' learning process by avoiding trial and error. LLF-Bench contains a diverse collection of tasks such as recommendation, constrained writing, navigation and robot control. LLF-Bench is designed to reflect an agent's learning and generalization capability, separating them from over-fitted performance on any given task. A key highlight is the configurable feedback system, classifying language feedback into performance, past behaviour explanations, and future suggestions — this encompasses existing RL environments, as well as imitation learning problems. Finally, we hope that LLF-Bench will serve as a research platform for developing and testing LLF agents, enabling the development of more general-purpose agents capable of learning to solve multiple tasks.

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

APPENDIX

# A ACCESSIBILITY CHECKLIST

1. Persistent URL: https://github.com/microsoft/LLF-Bench
2. Framework: We use the standard Gym API (Installation instructions and example codes are provided in the README.md of the github page.)
3. Long-term preservation: The project is hosted on a public repo on Github
4. License: MIT License
5. MetaData: On the project github page, we included a meta data table conforming to the stanford of schema.org

# B TASKS IN LLF-BENCH

LLF-Bench consists of 8 different problem sets, ranging from user-recommendation, poem-writing, navigation, to robot control. In the LLF setup, the reward is masked out (though the environments in LLF-Bench still return rewards for evaluation purposes). To solve these problem efficiently, an LLF agent needs to have sufficient common sense understanding of the natural language instruction and the feedback. In addition, the agent needs to be able to *learn* from environmental interactions and feedback. We intentionally design these suites of problems such that, while the agent can tell success from the instruction and the environmental observation, it is difficult for the agent to infer the optimal policy from them without additional learning.

These problem sets feature different action spaces, problem horizons, and test different abilities of LLF agents. We provide a summary in Table 5 and next describe each problem set in more detail.

| Problem Set | Action Space | Horizon | Stateful | Instruction | Feedback |
|---|---|---|---|---|---|
| llf-bandit | Discrete | 1 | No | b, p, c | all |
| llf-poem | Text | 1 | No | b | all |
| llf-reco-movie | Text | 1 | No | b, c | all |
| llf-optimization | Continuous | 10 | Yes | b | all |
| llf-parking | Continuous | 100 | Yes | b | r, hp, hn |
| llf-gridworld | Finite | 20 | Yes | b, p, c | all |
| llf-alfworld | Text | 100 | Yes | b | all |
| llf-metaworld | Continuous | 30 | Yes | b | r, hp, hn, fp |

Table 5: Properties of problem sets included in LLF-Bench. Instruction and Feedback column denote the types of instruction and feedback that are supported by the environment. If feedback is all, then it means that all 5 feedback (r, hn, hp, fn, and fp) are supported.

## B.1 LLF-BANDIT

llf-bandit is a verbalized version of the classic multi-armed bandit problem. We built llf-bandit based on gym-bandits[4] by adding natural language task instruction and feedback. There are a total of 8 bandit problems in llf-bandit. For each problem, the task instruction tells the task name from the underlying gym-bandits, that the goal is a bandit problem, as well as the feasible actions. While being a bandit problem, llf-bandit's feedback does not necessarily convey the reward value in text (it depends on the configuration of the feedback type). When reset, the environment randomizes the order of actions and, if applicable, the underlying reward function. The agent here needs to learn to explore and exploit in multiple rounds of interactions to find the best arm as fast as possible with small regret (measured in terms of the hidden rewards). Overall, llf-bandit tests the agent's learning ability in an unknown environment with a finite number of actions.

---

[4]MIT License

## B.2 LLF-POEM

`llf-poem` is a collection of text-generation tasks requiring a poem to be written that conforms to a particular number of lines and number of syllables for each line. Even though there are many types of formal poems, the current set of tasks supports basic types that follow syllable and line-based constraints. Such formal poems include Haiku (a three-line short poem following a 5-7-5 syllable pattern), Tanka (a five-line short poem following a 5-7-5-7-7 pattern), and custom environments where a user can specify the number of lines and how many syllables per line. We use the CMU Pronouncing Dictionary for syllable verification[5]. `llf-poem` provides detailed fine-grained feedback on each line – a good environment to test whether the LLM-based agents can improve quickly given feedback.

## B.3 LLF-RECO-MOVIE

`llf-reco-movie` is an environment that simulates user-recommendation system interactions on the topic of recommending movies. To simulate a user, the environment will first randomly sample a user preference profile over a set of attributes such as the type of entertainment (TV show or movie), year range (80s, 90s, 2000s, or recent), preferred genres (Action, Comedy, Documentary, etc.), and age restriction (child/family-friendly or R-rated). Then, a mask will be sampled to randomly hide one or more of the preferences in the initial request. An agent needs to recommend a few items (no restriction on the number of items) that all satisfy the stated preference. An item-by-item feedback is provided in this environment to point out detailed preference violations that can allow LLMs to improve their recommendations. The reward is defined as the percentage of recommended items being correct. $r \in [0, 1]$. This is a classic slate recommendation setup (Li et al., 2011; Swaminathan et al., 2017).

## B.4 LLF-OPTIMIZATION

`llf-optimization` provides an easy-to-use interface with automatic procedurally generated feedback that examines LLMs' ability to make a series of proposals $x$ to minimize a particular loss function $y = f(x)$. The feedback provided in this environment is created by computing gradient $\frac{dy}{dx}$ and then verbalizing this information based on the change in input between the previously proposed $x$ and the current chosen $x$. For each optimization problem, the input range is bounded: $x \in [x_{\min}, x_{\max}]$, and the reward is simply $\min(-y_t, -y_{\max})$ (to prevent any choice of $x$ that is outside of the bound). We provide implementations of 8 classic loss functions (Rosenbrock, Bohachevsky, etc.), and the base class is easily extendable to cover other loss functions. This is an environment where we can measure LLM's ability to make decisions with observed information on an unknown loss landscape.

## B.5 LLF-PARKING

`llf-parking` extends the Highway gym environment to LLF-Bench. It is a long-horizon goal-conditioned continuous control task where the agent can manipulate the throttle and steering input to an ego-vehicle. It must park the ego-vehicle in a given location without colliding with any obstacles in the environment. We extended the environment by (1) describing the observation and action spaces in text, and (2) verbalizing the per-time-step reward (distance to goal) to provide text feedback about goal progress and obstacle collisions. An agent must learn how its control inputs affect the vehicle's dynamics, and plan to accomplish the eventual parking goal.

## B.6 LLF-GRIDWORLD

The `llf-gridworld` domain models a navigation agent in a graph-based gridworld. The world is represented by a graph where rooms are denoted by nodes and edges denote doors. A room can have at most 4 doors along the north, south, east and west direction. These directions form the agent's action space. At any given time, the agent is in exactly one of the rooms. The agent's observation describes the current room including all the objects in it, and the different doors that are available. If the agent takes an action, such as $a = north$, then it will transition from its current room, to the

---

[5]http://www.speech.cs.cmu.edu/cgi-bin/cmudict

room connected by the door along the north direction, if one exists. If no such door exists, then the agent stays in the same room. All transitions are deterministic. A room can contain many different types of objects. A unique room, called the treasure room, contains the treasure object. The agent starts in a start room and its goal is to navigate to the treasure room. The number of rooms, objects, object type, and distance to the treasure, can be easily customized.

### B.7 `LLF-ALFWORLD`

The `llf-alfworld` environment is a wrapper built on top of the popular `AlfWorld` text-game environment[6] (Shridhar et al., 2021) which itself is built as a parallel to the embodied `Alfred` dataset (Shridhar et al., 2020). `llf-alfworld` contains multi-step reasoning tasks, where in each episode, the agent is given an instruction in a house setting and must take a sequence of actions to fulfill this instruction. In each step, the agent is given a textual description of what it sees and a list of valid actions. The agent generates a text action (e.g., *open drawer 1*), which if it is valid can change the agent's observation, and if it is invalid then results in no change. The agent additionally gets a reward for each action. The goal of the agent is to maximize the total reward by solving the task. Unlike the `llf-gridworld` setting, the agent is tested for generalization as each episode can contain a new task in a possibly new house environment. The main addition in `llf-alfworld` is the capability to provide text feedback instead of reward. The text feedback is generated using an optimal trajectory for that episode, as well as the instantaneous reward and the list of valid commands for each time step.

### B.8 `LLF-METAWORLD`

`llf-metaworld` is based on the existing Meta-World v2 benchmark[7] (Yu et al., 2019) and supports both text (of low-dimensional states) and image observations. Meta-World consists of 50 simulated robotic manipulation tasks, in each of which a robotic Sawyer arm needs to move an object into a specified position, e.g., push a puck to a goal location or press a button. An agent trying to accomplish an `llf-metaworld` task is presented with an instruction stating that the task is about getting a robotic manipulator to successfully handle an object and explaining what each dimension of the agent's 4D state space means. By default, the environment interprets an agent's action as a target pose where the arm should move[8], and tries to move the arm there using Meta-World's built-in P-controller. At each time step, the agent receives as observation a description of the current state mentioning the pose of the arm and all relevant objects in the scene. The language feedback here may include advice on where to move the arm next and where not to move it.

## C  GYM INTERFACE OF LLF-BENCH

LLF-Bench formalizes a wide variety of decision-making problems by extending the popular OpenAI Gym API. The API contains three key functions — `make`, `reset`, `step` — that are semantically similar to their Gym namesakes and detailed below. A sample code snippet for interaction with LLF-Bench's Gym interface can be found in Figure 3.

- `make`: Returns an *Environment* object similar to `gym.make`. An LLF-Bench *Environment* extends classic Gym Environments (e.g., with well-defined `ActionSpace` and `ObservationSpace`) with two additional concepts, `instruction` and `feedback`, that are explained below.

- `reset`: After an environment is initialized using `make`, it should be `reset` to receive the initial *Observation* from the *Environment*. LLF-Bench *Observation* is a Python dictionary containing `gym.Observation` (i.e., an observation that is contained in the

---

[6]MIT License

[7]MIT License

[8]The dynamics of `llf-metaworld` differs from the one in the original Meta-World. Here the agent controls the target location (the simulator runs the P-controller to act in the original Meta-World environment for several steps until the target location is reached or it is timed out), whereas in the original environment the agent controls force to incrementally change the end-effector. This design is to make the problem horizon shorter and more closely mimic the common use cases of industrial robotic manipulators.

```python
import llfbench as gym

# Environments in the benchmark are registered following
# the naming convention of llf-*
env = gym.make('llf-gridworld-v0')

done = False
cumulative_reward = 0.

# First observation is acquired by resetting the environment
observation = env.reset()

while not done:

    # Observation is dict having 'observation', 'instruction', 'feedback'
    # Here we print the observation and ask the user for an action
    action = input( observation['observation'] + '\n' +
                    observation['instruction'] + '\n' +
                    observation['feedback'] + '\n' +
                    'Action: ' )

    # Gridworld has a text action space, so TextWrapper is not needed
    # to parse a valid action from the input string
    observation, reward, terminated, truncated, info = env.step(action)

    # reward is never revealed to the agent; only used for evaluation
    cumulative_reward += reward

    # terminated and truncated follow the same semantics as in Gymnasium
    done = terminated or truncated

print(f'Episode reward: {cumulative_reward}')
```

Figure 3: Sample Python code snippet for interacting with LLF-Bench environments.

environment.ObservationSpace) as well as instruction and feedback keys. If the environment uses randomization, then the random number generator can be seeded with the seed parameter as input.

- step: Takes as input an action that is contained in the environment.ActionSpace, and returns a LLF-Bench *Observation* dictionary which includes the instruction and feedback keys. In addition to the *Observation*, step also returns scalar *reward*, boolean flags *truncated* and *terminated* and a miscellaneous *info* dictionary which have the same semantics as Gymnasium environments. An agent for LLF-Bench is expected to solve tasks using the feedback contained within *Observation*, **without** using the *reward* signal. Signals like *reward* and *info* are provided for backward compatibility with Gymnasium and for automated evaluation.

Note that under the hood, LLF-Bench implements all *Environment* objects as compatible with the Gymnasium standard. We provide EnvironmentCompatibility wrappers if the *Environment* is instead otherwise compatible with the deprecated Gym (pre-0.21 version) standard. We similarly include TextWrapper wrappers that can convert any LLF-Bench *Environment* with bespoke ObservationSpace and ActionSpace into one with text as the observation and action spaces. This wrapper allows one to directly interface LLM-based agents with LLF-Bench environments and assess their learning and decision-making behavior.

Although each step also returns a scalar *reward*, the convention we follow (and recommend to users of LLF-Bench) is that the agent never sees the reward. It can only access the information in *observation*, *instruction* and *feedback* to decide its actions (e.g., see line 17 in Figure 3).

## C.1 INSTRUCTION AND FEEDBACK

*Instruction* is a string that is defined inside the *Environment* and describes in natural language the problem that a decision-maker must solve. We recommend that agent-designers should not inspect and overfit to a specific instruction describing the desired task in an environment; the default behavior of LLF-Bench environments is to paraphrase instructions in different ways to minimize the chances of prompt overfitting. Three different types of *Instruction* are supported in LLF-Bench, and can be toggled by passing an appropriate `instruction_type` to the `make` command of a LLF-Bench environment:

- Basic: `instruction_type = 'b'`. This is the default instruction type for LLF-Bench environments. The instructions provide an agent with the goal, semantics of its action space, as well as the expected syntax of its responses. The instruction provides enough information for a competent agent (e.g., a literate human) to begin interacting with the environment.
- Complete: `instruction_type = 'c'`. The instructions additionally provide information to reliably infer (e.g., by a literate human) an optimal policy for achieving the goal.
- Practical: `instruction_type = 'p'`. It contains the *Basic* instructions, and additionally includes *Feedback* for previously executed actions. The goal of a learning agent is to infer the optimal policy (i.e., comparable in performance to the one with `instruction_type = 'c'`) as quickly as possible.

*Feedback* is a string that provides the signal for an agent to learn from its interaction. LLF-Bench implements two kinds of feedback: an atomic feedback, and a composite feedback. The type of feedback an environment provides to an agent is set by passing an appropriate `feedback_type` parameter to `make`. Atomic feedbacks are inspired by the education research literature (Shute, 2008). LLF-Bench currently supports 5 different types and we plan to include new styles (to include e.g., questioning) in the future:

- `feedback_type = 'r'`: This is the textualization of the scalar reward signal or success signal from classical RL. By using the text-wrapper and this feedback type, several classical RL environments (implemented in OpenAI Gym or Gymnasium) can be comparably tested with LLF agents in LLF-Bench.
- `feedback_type = 'hp'`: This *hindsight positive* feedback provides an explanation about a past action by the agent that was desirable.
- `feedback_type = 'hn'`: This *hindsight negative* feedback provides an explanation about a past action by the agent that was undesirable.
- `feedback_type = 'fp'`: This *future positive* feedback provides a suggestion for a potential future action that could be desirable.
- `feedback_type = 'fn'`: This *future negative* feedback provides a suggestion for potential future actions that should be avoided.

`feedback_type = 'r'` corresponds to the **current performance** evaluation from the education research literature, whereas `feedback_type = 'fp'`, `'fn'` correspond to **future behavior** suggestion. Finally, `feedback_type = 'hp'`, `'hn'` correspond to the **past behavior** explanation style of feedback studied in the education research literature.

Composite feedback types allow the environment to provide the agent multiple kinds of atomic feedbacks. This makes for a more realistic learning problem, rather than the same type of atomic feedback at every `step` of the environment.

- `feedback_type = 'a'`: *All* of the Atomic feedback types that are supported by the environment are provided to the agent at each round of interaction.
- `feedback_type = 'm'`: The agent receives a *Mix* of different atomic feedbacks. A random subset of the supported feedback types are sampled by LLF-Bench to provide to the agent at each step.
- `feedback_type = 'n'`: The agent receives *No* feedback, this mode is provided for debugging purposes.

The `make` API accepts any of the composite feedback types, or any subset of the atomic feedback types to allow fine-grained control of the learning signal that an agent can receive from LLF-Bench environments. The default behavior in `make` for any environment uses `feedback_type = 'a'`.

## C.2 INSTRUCTION AND FEEDBACK RANDOMIZATION

To reduce the sensitivity of learning agents to a specific text realization, LLF-Bench implements a template-based paraphrasing system, by which users can randomize the instruction and the feedback that the agent receives. For each problem in LLF-Bench, we implement about 4-20 paraphrased templates for each instruction and each feedback type. When the randomization options are turned on, the LLF-Bench environment will randomly choose one from these curated templates to formulate the language instruction and feedback returned to the agent. LLF-Bench also provides the option to deterministically use a particular template. The randomness of paraphrasing can be controlled by setting the `seed` parameter in the OpenAI Gym `reset` function.

Compared with using language models to generate feedback on-the-fly, the use of templates offers advantages: (1) the latter is free, while the former can be very expensive. (2) The latter results in far higher reproducibility: it is very hard or even impossible to guarantee that an LLM produces exactly the same output (gives the same feedback) for the same prompt (for the same state). (3) Some generic templates can be broadly useful across many environments, serving as an initial example when building a new environment. All three reasons are very important for a benchmark, and in general the use of synthetic data has been common in the literature (Hudson & Manning, 2019; Hermann et al., 2020; Blukis et al., 2018), although we acknowledge that it has its own drawbacks.

# D RELATED WORK

**Grounded Language Learning** Reinforcement learning with textual information has been studied under the branch of multi-modal representation learning. This branch of study has several focuses that are both similar and different from our goal with LLF-Bench. One focus deals with ambiguity and difficulty in understanding instructions or goals specified by natural language (Wang et al., 2016; Bahdanau et al., 2019; Chaplot et al., 2018). While the ambiguity of instructions is a concern, we focus more on robustly behaving under different instructions that all represent the same underlying goal. Another focus of this body of work is to ground visual information with textual instruction – a core aim of multi-modal representation learning (Bisk et al., 2016; Misra et al., 2017), with an extension to robotic interaction (Karamcheti et al., 2022; 2023). Language provides a natural shared representation that enables easier transfer between different tasks (Hanjie et al., 2021) or supplies important information such as safety constraints for a policy (Yang et al., 2021). In previous work, feedback is often not considered. When feedback is considered, it is usually framed as error messages from a syntax parser (if the action space is text) and can indeed be incorporated into learning (Côté et al., 2019). This type of feedback corresponds to `feedback_type = 'hn'` in our setup.

**Text-based Games** Extending from using reinforcement learning for solving complex games, there are many text-based games that include challenges such as the navigation of space, manipulation of the environment to achieve goals, and reaction to random events. Narasimhan et al. (2015) repurposed a classic text adventure game, Zork, where both observation and action space are text. Côté et al. (2019) proposed a set of text-based game environments and included a few carefully designed challenges for RL to solve, such as large state and action space (determined by the vocabulary size) and long credit assignment. On the other spectrum, Küttler et al. (2020) created a learning environment from the game NetHack. Although the game state is represented with hundreds of text symbols, policy learning is conducted on the screenshot of the terminal. Similarly, BabyAI (Chevalier-Boisvert et al., 2019) is a set of procedurally generated grid-like maze environments – the objects and representation in the environment are a fixed set of symbols. None of these environments consider providing language feedback on the agent's action.

**Learning from Language Feedback** Providing feedback to an RL agent's action as part of the learning signal beyond task rewards has been studied in robotics. However, most of the efforts were limited to eliciting binary preference feedback (Sadigh et al., 2017; Biyik & Sadigh, 2018) or

ranking-based feedback from real people (Basu et al., 2019). Sumers et al. (2021) crowd-sourced a small feedback dataset on a small game. They considered three types of feedback, evaluative feedback (which corresponds to `feedback_type = 'r'`), descriptive feedback (which in our setup is decomposed into `feedback_type = 'hp'`, `'hn'`), and imperative feedback (which corresponds to `feedback_type = 'fp'`, `'fn'`). They then used a sentiment classifier to extract coarse information from this feedback to improve the policy's behavior. Nguyen et al. (2021) proposed an approach to map textual instructions to trajectories in embodied settings by assuming that a user can label a generated trajectory with the instruction that is likely to generate the trajectory under the optimal policy. More recently, Cui et al. (2023); Liu et al. (2023a) studied the case of language feedback as corrections to a robotic arm at any time of the task execution, which is an instance of the LLF setup that we are considering.

**LLM Sensitivity to Prompts** A long line of work has investigated smaller-scale language-based systems' sensitivity to different expressions that have the same underlying meaning. They can be categorized as adversarial attacks to text-based systems (Ribeiro et al., 2018; Wallace et al., 2019) or as mechanisms to improve language-based systems' output via self-consistency (Edunov et al., 2018). More recently, the lack of robustness to prompts has been found on large language models as well (Liu et al., 2023c; Wolf et al., 2023). Zhu et al. (2023) proposed a benchmark dataset to investigate the robustness of LLMs on different types of prompts that can contain user errors for tasks related to natural language.

# E EXPERIMENT DETAILS

We use the TEXTWRAPPER provided with LLF-Bench to format observations and feedback into text, suitable for evaluating LLMs as agents. Then we implemented a Reflexion-based basic agent (Shinn et al., 2023) that formats up to 20 most recent observation-action-feedback tuples into an LLM's context. Its system prompt is listed Figure 4 and its user prompt Figure 5 (For Meta-world tasks with image observations, we add an additional instruction to let it reflect on the observation before making the decision.) We extract the action from the LLM's output using template matching starting with "Response: ' and feedback the extracted action to the LLF-Bench environment's Gym interface. When error happens due to LLM outputting inadmissible actions, we catch the error and send it back as feedback to the agent; we found that LLMs often are able to understand such error feedback and use them to correct the action format in the next round.

All environments are initialized with horizon of $H = 30$ and are stopped if success earlier if the agent successfully solves the problem (e.g., in Meta-World problems). That is, the RESET function of the environment is called after at most 30 time-steps to initiate a new episode. We reset the agent at the start of each episode and compute the statistics by 50 independent episodes (using seed 0).[9] All experiments are run with the basic instruction of LLF-Bench (see Appendix C). Note that in the experiments, the agents do not see the numerical reward as feedback necessarily, while the language feedback might contain some information of the instantaneous performance. We conducted all experiments using API access to SoTA LLMs queried during the month of May 2024 on a single computer with Intel(R) Core(TM) i9-9980XE CPU @ 3.00GHz and 64GB of memory.

## E.1 COMPARISON BETWEEN BASIC AGENT AND ORIGINAL REFLEXION AGENT

Our Reflexion-based agent, denoted as Basic Agent, differs from the original Reflexion agent in that the original Reflexion agent implementation additionally stores the reflections in the agent's memory buffer. We also implemented and run the original Reflexion agent (denoted as Reflexion Agent below), and compared it with the Basic Agent we used in the paper on the challenging LLF-Meta-World tasks. The results are shown below in Figure 6 and Figure 7. We observe that Basic Agent, despite being simpler, performs better across different tasks.

---

[9]Since there is no additional training runs, using a single seed with multiple independent evaluation episodes is equivalent to using multiple seeds with one evaluation each.

```
1 You are an agent tasked to solve an interactive problem with verbal
      feedback. You will see an Instruction. After you choose an action,
      you will see the feedback from the environment. Your goal is to
      choose the right actions to solve the task as fast as possible,
      according to the Instruction.
2
3 Answer in the following format: First, begin with "Thought:" and write
      down your reflection on the feedback. Then in the next line write
      your response beginning with "Response:" and provide your chosen
      action. ONLY provide the chosen action after "Response:", without any
       additional comments or thoughts. Anything extra will cause errors,
      as your responses will be parsed by a computer program, not a human.
4
5 Here is an example for an Instruction which asks you to choose a number
      between 1 and 10:
6
7 Thought: I should choose a number that is not too high or too low, so I
      will choose 5.
8 Response: 5
9
10 An invalid response would be:
11
12 Thought: I should choose a number that is not too high or too low, so I
      will choose 5.
13 Response: I choose number 5
```

Figure 4: System prompt used for all LLMs.

```
1 History of feedbacks: {history}
2
3 Current observation: {observation}
4
5 Instruction: {instruction}
```
Listing 1: User Prompt for All Problems, except Meta-World with Image Observation

```
1 History of feedbacks: {history}
2
3 Current observation: {observation}
4
5 Instruction: {instruction}
6
7 Change of reply format: The new reply format is Observation, Thought, and
       Response. Write down what you see in the image in Observation
      section, and in Thought reflect on the feedback as well as
      Observation.
```
Listing 2: User Prompt for Meta-World Problems with Image Observation

Figure 5: User prompts used for all LLMs.

## F  EXAMPLE INSTRUCTION, OBSERVATION, AND FEEDBACK

This section provides some examples of instruction, observation, and feedback of environments in LLF-Bench. For environments without observation, we do not list the **Example Observation** below. Also, for compactness, we remove the text formatting (such as spacing and indentation) of instruction, observation, and feedback. Please refer to the code for the exact text presentation given to the agent. Example image observation of `llf-metaworld` environments can be found in Figure 1.

| Problem sets and problems | Basic Agent | Reflexion Agent |
|---|---|---|
| reach | **1.00** (0.00) | **1.00** (0.00) |
| button-press-wall | 0.82 (0.05) | **0.86** (0.05) |
| bin-picking | **0.88** (0.05) | 0.68 (0.07) |
| pick-place | **0.68** (0.07) | **0.68** (0.07) |
| assembly | **0.00** (0.00) | **0.00** (0.00) |
| push | **0.86** (0.05) | **0.86** (0.05) |
| box-close | **0.88** (0.05) | 0.64 (0.05) |
| hand-insert | 0.16 (0.05) | **0.28** (0.06) |
| faucet-open | **1.00** (0.00) | **1.00** (0.00) |
| dial-turn | **1.00** (0.00) | **1.00** (0.00) |

Figure 6: Comparison between Basic Agent and Reflexion Agent on `llf-metaworld` tasks **with all feedback types**. Both agents use GPT-4o. Table shows mean and standard error of return, computed with 50 episodes.

| Problem sets and problems | Basic Agent | Reflexion Agent |
|---|---|---|
| reach | **0.82** (0.04) | 0.58 (0.07) |
| button-press-wall | **0.56** (0.05) | 0.46 (0.07) |
| bin-picking | **0.00** (0.00) | **0.00** (0.00) |
| pick-place | **0.46** (0.05) | 0.10 (0.04) |
| assembly | **0.00** (0.00) | **0.00** (0.00) |
| push | **0.35** (0.05) | 0.28 (0.06) |
| box-close | **0.00** (0.00) | **0.00** (0.00) |
| hand-insert | **0.13** (0.03) | 0.06 (0.03) |
| faucet-open | 0.57 (0.05) | **0.78** (0.06) |
| dial-turn | **0.05** (0.02) | 0.04 (0.03) |

Figure 7: Comparison between Basic Agent and Reflexion Agent on `llf-metaworld` tasks **with Reward and Hindsight feedback types only**. Both agents use GPT-4o. The table shows mean and standard error of return, computed with 50 episodes. Note that the performance drops markedly compared to using all feedback types (Figure 6).

### F.1 LLF-RECO-MOVIE-V0

**Example Instruction** You are a helpful assistant trying to recommend movies or tv shows to your users according to what they want. Sometimes, your users don't fully tell you their preferences at the start, but once you make recommendations, they will tell you truthfully what they like and don't like. Please produce a valid json list with a dictionary: ["title": "movie1", "title": "movie2"] Example Hit me with your best Western movie suggestions from the 2000s or 80s. Please point me in the right direction.

**Example Feedback** I can find all the recommendations online, nice! Your recommended picks are movies, wonderful! The recommendations span a broader range than just Western movies. The recommendations are not from the 2000s or 80s. The recommendations are all child-friendly, awesome! Indeed, these recommendations are categorized as Western: Unforgiven is Drama or Western, True, these recommendations are from the 2000s or 80s: No Country for Old Men is from 2007, Silverado is from 1985, 3:10 to Yuma is from 2007, It turns out that these recommendations are not Western: No Country for Old Men is Crime, Drama, or Thriller, Silverado is Action, Crime, or Drama, 3:10 to Yuma is Action, Crime, or Drama, These recommendations are not from the 2000s or 80s: Unforgiven is from 1992, Make recommendations that are Western, like True Grit and The Good, the Bad and the Ugly. Identify movies that were released during 2000s or 80s, like Pearl Harbor and Black Hawk Down. Do not make recommendations that are not Western, not like L.A. Confidential or The Addams Family. Do not make recommendations that are not from 2000s or 80s, like True Grit or L.A. Confidential.

## F.2 LLF-OPTIMIZATION-MCCORMICK-V0

**Example Instruction**    You are trying to minimize the output (y) of a function by choosing input (x). The goal is to choose x such that y is as small as possible. You get to observe y once you choose the value of x, where x is a 2-dimensional vector. This means x = [x1, x2], where x1 and x2 are real numbers. The range of x1 and x2 is [-1.5, 4]. Please do not choose x outside of this range. Choose x within 10 attempts. You can choose to stop at any time. Output format: x = [x1, x2]

**Example Observation**    x=[-1.2746, -1.4091] Function outputs y = -1.034818410873413 You have 10 attempts left! Please output the next x that will make this function output the smallest y. Format: x = [x1, x2]

**Example Feedback**    You are yet to achieve the minimum! You selected x = [-1.2, -1.5] from [-1.3 -1.4]. Correctly, increasing the first number in -1.3 contributes to minimizing y. Your decision was x = [-1.2, -1.5] from [-1.3 -1.4], and it's accurate that Decreasing the second number in -1.4 aids in minimizing y. Your pick was x = [-1.2, -1.5]. Minimize y by issuing a larger number than the first element of -1.2. You went with x = [-1.2, -1.5]. Now, aim to output a smaller number than what is at the start of -1.5 to achieve a smaller y. You decided on x = [-1.2, -1.5]. Avoid producing a smaller number than the first number in -1.2 if you want to minimize y.Your selection fell on x = [-1.2, -1.5]. For the purpose of minimizing y, do not provide a larger number than the second from -1.5. Thought: From the feedback, it seems that further decreasing the first element and increasing the second element from [-1.2746, -1.4091] did not minimize y. I should try increasing the first element and decreasing the second element within the given range to see if the function's output decreases further.

## F.3 LLF-GRIDWORLD-V0

**Example Instruction**    You are in a house that has multiple rooms. When you are in a room, you can see all the objects that this room contains but cannot see objects in different room. At a given time, you can only be in one room. Each room can have a door along the North, South, East and West direction. Different rooms can different number of doors. You can follow a direction to go from one room to another, provided there is a door in that direction. If there is no door along that direction, then you will remain where you are. You will start in a room. Your goal is to navigate to the unique room which has the treasure. You have an action space of size 4. Action 0 leads to going North. Action 1 leads to going East. Action 2 leads going west. Action 3 leads to going South.

**Example Observation**    You are in lobby-1 room. You have a door to the south of you that takes you to the toilet-1 room. You have a door to the west of you that takes you to the drawing room-3 room. You have a door to the east of you that takes you to the corridor-3 room.

**Example Feedback**    You got a reward of 0.0. You did the right thing by following the south direction in lobby-1. You were right in not going in the west direction in your latest move. Now that you are in toilet-1, make sure to follow the east direction. You should not follow the west direction in this toilet-1.

## F.4 LLF-HIGHWAY-PARKING-V0

**Example Instruction**    Your goal is to control a vehicle to park in a desired location, while ensuring that it does not collide with any obstacles or other vehicles. You will receive the observation of the vehicle's state as well as the desired parking location represented by an array of numbers. The dimensions of the array correspond to [x, y, vx, vy, cos_h, sin_h]. That is, the first 2 dimensions denote the position, the next 2 denote the velocity, and the last 2 denote the orientation. Your action is a 2-dim vector, where the first dimension controls the throttle input, and the last dimension controls the steering input. Throttle is a number between -5 and 5, representing acceleration in units of m/s$^2$. Steering is a number between -pi/4 and pi/4, representing the steering angle in radians. Present a correct action in the form of [throttle input, steering input].

**Example Observation** OrderedDict([('observation', array([-0.10405679, -0.04218505, -2.68598268, -1.33622492, 0.89532756, 0.44540831])), ('achieved_goal', array([-0.10405679, -0.04218505, -2.68598268, -1.33622492, 0.89532756, 0.44540831])), ('desired_goal', array([2.000000e-02, 1.400000e-01, 0.000000e+00, 0.000000e+00, 6.123234e-17, 1.000000e+00]))])

**Example Feedback** The reward is -0.4557528810103727

### F.5 LLF-POEM-LINESYLLABLECONSTRAINEDPOEM-V0

**Example Instruction** Are you competent to write a poem for me? It should be 4 lines long and the syllable count for each line should match a 10-10-5-8 pattern.

**Example Feedback** The poem that was assembled is not right. The lines are correct because they follow the right syllable count: line 1 has 10 syllables,line 2 has 10 syllables,line 4 has 8 syllables. Poem must contain exactly 10-10-5-8 syllables across 4 lines, but line 3 does not. Here are some pointers to help you resolve your error: The sentence: "Gentle whispers," has 4 syllables, not the 5 syllables it should have. You need to revise the sentence to have more syllables.

### F.6 LLF-BANDITS-BANDITTENARMEDGAUSSIAN-V0

**Example Instruction** 10 armed bandit mentioned on page 30 of Sutton and Barto's [Reinforcement Learning: An Introduction] https://www.dropbox.com/s/b3psxv2r0ccmf80/book2015oct.pdf?dl=0) Actions always pay out Mean of payout is pulled from a normal distribution (0, 1) (called q*(a)) Actual reward is drawn from a normal distribution (q*(a), 1) Find the best action as fast as possible. Your action is an integer between 0 and 9.

**Example Feedback** You've been rewarded with 2.041880926155133. This arm isn't the best because it doesn't offer the highest expected reward. If you decide on action 7, you'll be rewarded with an expected 1.4102046311312142. Observation: The action 4 is not the best choice as it results in an expected reward of -0.18158257273119596

### F.7 LLF-ALFWORLD-V0

**Example Instruction** You are in a house with a variety of objects. Your task is to: find two saltshaker and put them in sidetable. You have to take a sequence of actions to full fill it. When you take an action, you can change the world. You will be told at each step, what actions are allowed and you must pick only one of those actions.

**Example Observation** -= Welcome to TextWorld, ALFRED! =-

You are in the middle of a room. Looking quickly around you, you see a cabinet 4, a cabinet 3, a cabinet 2, a cabinet 1, a coffeemachine 1, a countertop 1, a diningtable 3, a diningtable 2, a diningtable 1, a drawer 1, a fridge 1, a garbagecan 1, a microwave 1, a sidetable 1, a sinkbasin 1, a stoveburner 4, a stoveburner 3, a stoveburner 2, a stoveburner 1, and a toaster 1.

Your task is to: find two saltshaker and put them in sidetable.. You are allowed to take the following actions: go to cabinet 1, go to cabinet 2, go to cabinet 3, go to cabinet 4, go to coffeemachine 1, go to countertop 1, go to diningtable 1, go to diningtable 2, go to diningtable 3, go to drawer 1, go to fridge 1, go to garbagecan 1, go to microwave 1, go to sidetable 1, go to sinkbasin 1, go to stoveburner 1, go to stoveburner 2, go to stoveburner 3, go to stoveburner 4, go to toaster 1, inventory, look.

**Example Feedback** Your latest action gives you a reward of 0. The action look, is what you should have chosen in your last move. At the last step, you did not take the action go to microwave 1, and this was a good thing as it was a bad action. The optimal action to take in the next step is open cabinet 1. You should avoid the action go to sidetable 1 in the next step. Thought: To find the saltshakers, it's logical to start searching places where kitchen items are typically stored. Cabinets are a good starting point.

## F.8 LLF-METAWORLD-BUTTON-PRESS-WALL-V2

**Example Instruction**  Your job is to control a Sawyer robot arm to solve a button-press-wall task. You will get observations of the robot state and the world state in the form of json strings. Your objective is to provide control inputs to the robot to achieve the task's goal state over multiple time steps. Your actions are 4-dim vectors, where the first 3 dimensions control the movement of the robot's end effector in the x, y, and z directions, and the last dimension controls the gripper state (0 means opening it, and 1 means closing it). You action at each step sets the robot's target pose for that step in absolute coordinate. The robot will move towards that pose using a P controller.

**Example Observation**  "hand_pos": "[0.012 0.561 0.138]", "hand_closed": "0.287", "button_pos": "[-0.016 0.687 0.115]" Action: [-0.02, 0.56, 0.3, 1.0]

**Example Feedback**  You've received a reward of 0.3445458270735896. You are making progress towards achieving the goal. Keep it up! The target [-0.02 0.57 0.3 1. ] is promising.

