# OpenReview forum: "LLF-Bench: A Benchmark for Interactive Learning from Language Feedback"
_ICLR.cc/2025/Conference — Submitted to ICLR 2025_

### Official Review · Reviewer_QDaS · 2024-11-01

**Soundness:** 2
**Presentation:** 3
**Contribution:** 3
**Rating:** 5
**Confidence:** 4

**Summary:**

The authors introduce LLF-Bench, a benchmark for evaluating the ability of AI agents in Learning from Language Feedback (LLF). LLF-Bench provides a suite of 8 diverse tasks including movie recommendation, poem writing, numerical optimization, and robot control. Unlike traditional RL gym environments, LLF-Bench environments provide no numerical reward, instead providing a natural language instruction at the beginning, and natural language feedback to the agent at each step. Language feedback is generated via manually written feedback templates for consistency and efficiency, while ensuring a diverse set of paraphrased templates to prevent overfitting.

**Strengths:**

- LLF-Bench provides a useful benchmark and research platform for LLF, a very interesting contemporary topic which has lacked systematic environments for evaluation.
- Good conceptual discussion of how this compares to standard RL and why LFF has advantages over RL.
- Compatibility with existing OpenAI Gym environments is a clever design choice that makes this easily extensible.
- Useful distinction of 3 different types of feedback: Reward, Future Feedback, Hindsight Feedback
- Use of templates (as opposed to LLM-generated feedback) is a sensible choice for reproducibility and cost/compute efficiency.

**Weaknesses:**

My biggest complaint is insufficient experiments and analysis on the benchmark.
- In my mind, LLF-Bench needs to provide evidence justifying itself as a meaningful improvement over simply passing in standard RL gym observations and rewards into a language model. The paper provides good motivation for why we should expect language feedback to be more efficient that traditional rewards, but fails to provide evidence for this hypothesis.
- Relatedly, LLF-Bench introduces 3 interestingly different kinds of feedback types (Reward VS Future VS Hindsight), but do little to show why they are useful. In theory, comparing the setups between Table 2 and Table 3 gives us an ablation of how removing Future Feedback affects performance, but the separate presentation of the tables makes it very difficult to compare, and the text does not discuss the findings as far as I can see.
- The authors also don't clearly motivate why they run the specific experiments of Table 2 and Table 3. Table 2 makes sense as a default / main setup, but ablations should be clearly motivated and state what questions they seek to answer. For example, I would be interested to see comparisons of: [Reward] vs [Reward + Hindsight] vs [Reward + Hindsight + Future] or similar experiments trying to isolate the impact of each feedback type.
- It would be useful to also compare LLM results against RL-trained agents using rewards as baselines, since that would allow us to compare how quickly RL agents learn from reward VS LLMs with language feedback. This would provide some useful investigation into claims that feedback is more efficient that RL rewards.

**Questions:**

- Something else I was looking for was further elaboration on which kinds of environments stand to benefit most from natural language feedback. The paper alludes to the idea that some tasks benefit more from language feedback:
> "real-life use cases, where a user needs LLM-based agents to handle tasks whose solution cannot be directly inferred from the task description and has to be learned from interactions and feedback instead (such as “make the title text larger” or “wrap the code with an error-catching block.”)."

  But it is not clear how the selected environments and the constructed feedback provide these interesting properties. Would be curious if there was more careful thought in the selection of these particular environments.

- Is there an error in Appendix F.2? "Example Feedback" includes a "Thought: From the feedback, it seems that..."

---

> ### Author Response · Authors · 2024-11-23
>
> Thank you for your thoughtful review. We address your concerns point by point below:
>
> > The paper provides good motivation for why we should expect language feedback to be more efficient than traditional rewards, but fails to provide evidence for this hypothesis.
>
> Large Language Models have been fine-tuned to follow instructions. Feedback, especially future positive (FP) or future negative (FN) can be seen as a form of instruction (e.g., “recommend an action movie” or “do not recommend a horror movie”). In Table 2 and 3, we already showed that by providing reward and different kinds of feedback types, there is a large variation in performance differences between LLMs. We highlight the comparison in the Movie-Rec env below as an example.
>
> | Movie-Rec | GPT-4o | llama3-70b | gemini-pro | phi3-mini-128k |
> |------|------|------|------|------|
> | Reward + Hindsight | -5.28 (1.55) |  -9.61 (1.73)  | -18.74 (1.77)  | -18.36 (1.97) |
> |Reward + All Feedbacks | -6.98 (1.08) | -6.10 (1.49) | -3.45 (1.13)  |-14.23 (1.91) |
>
>
> We can see this type of difference for all LLM models across all of our environment in Table 2 and 3, showing a sensitivity to the type of feedback it receives. We believe this motivates the need to investigate different feedback types.
>
> > Table 2 and Table 3 gives us an ablation of how removing Future Feedback affects performance, but the separate presentation of the tables makes it very difficult to compare, and the text does not discuss the findings as far as I can see.
>
> Thank you for bringing up this great point. We will add a new plot that contrasts the performance differences caused by different feedback types (Reward+Hindsight+Future, vs Reward+Hindsight, vs Reward) in the revision. (See the table below for the comparison).

---

> > ### Author Response · Authors · 2024-11-23
> >
> > > For example, I would be interested to see comparisons of: [Reward] vs [Reward + Hindsight] vs [Reward + Hindsight + Future] or similar experiments trying to isolate the impact of each feedback type.
> >
> > Per your request, we conducted new experiments using just the Reward feedback with gpt4o. We put them by the results of Table 1 and 2 for comparison. We updated our evaluation for the bandit environment to differentiate different policies better. The results are shown below:
> >
> > | EnvName | Reward, Hindsight, Future | Reward, Hindsight | Reward|
> > |----------------------------|-----------------------------------------|-----------------------------------------|-----------------------------------------|
> > | *10ArmedUniform*           | **-1.90** (0.04)                        | -5.01 (0.29)                            | -13.54 (1.81)                            |
> > | *10ArmedGaussian*          | **-1.96** (0.08)                        | -5.90 (0.31)                            | -16.22 (1.82)                            |
> > | *Booth*                    | **-3.93** (0.68)                        | -14.26 (1.81)                           | -29.62 (4.80)                           |
> > | *McCormick*                | **-0.19** (0.05)                        | -0.26 (0.03)                            | -0.27 (0.11)                            |
> > | *Rosenbrock*               | **-1.19** (0.33)                        | -10.51 (3.55)                           | -9.45 (3.42)                            |
> > | *SixHumpCamel*             | **-0.23** (0.06)                        | -0.56 (0.07)                            | -0.31 (0.11)                            |
> > | *reco-movie*               | **-5.28** (1.55)                        | -6.98 (1.08)                            | -11.74 (1.77)                           |
> > | *parking*                  | -14.32 (0.39)                           | -14.32 (0.28)                           | **-13.94** (0.43)                       |
> > | *Haiku*                    | -6.59 (1.51)                            | **-3.98** (0.82)                        | -12.52 (2.08)                           |
> > | *Tanka*                    | -9.03 (1.49)                            | -20.68 (1.35)                           | **-6.55** (1.74)                        |
> > | *LineSylConstr*            | **-13.92** (1.33)                       | -23.97 (0.64)                           | -24.64 (0.83)                           |
> > | *gridworld*                | **1.00** (0.00)                         | 0.95 (0.02)                             | 0.50 (0.07)                             |
> > | *alfworld*                 | **0.80** (0.06)                         | 0.64 (0.07)                             | 0.56 (0.07)                             |
> > | *reach*                    | **1.00** (0.00)                         | 0.82 (0.04)                             | 0.96 (0.03)                             |
> > | *button-press-wall*        | **0.82** (0.05)                         | 0.56 (0.05)                             | 0.36 (0.07)                             |
> > | *bin-picking*              | **0.88** (0.05)                         | 0.00 (0.00)                             | 0.00 (0.00)                             |
> > | *pick-place*               | **0.68** (0.07)                         | 0.46 (0.05)                             | 0.20 (0.06)                             |
> > | *assembly*                 | **0.00** (0.00)                         | **0.00** (0.00)                         | **0.00** (0.00)                         |
> > | *push*                     | **0.86** (0.05)                         | 0.35 (0.05)                             | 0.14 (0.05)                             |
> > | *box-close*                | **0.88** (0.05)                         | 0.00 (0.00)                             | 0.00 (0.00)                             |
> > | *hand-insert*              | **0.16** (0.05)                         | 0.13 (0.03)                             | 0.08 (0.04)                             |
> > | *faucet-open*              | **1.00** (0.00)                         | 0.57 (0.05)                             | 0.54 (0.07)                             |
> > | *dial-turn*                | **1.00** (0.00)                         | 0.05 (0.02)                             | 0.10 (0.04)                             |

---

> > > ### Author Response · Authors · 2024-11-23
> > >
> > > > It would be useful to also compare LLM results against RL-trained agents using rewards as baselines, since that would allow us to compare how quickly RL agents learn from reward VS LLMs with language feedback.
> > >
> > > Thank you for the suggestion! For Meta-World [1], see Figure 12, 13, 14, an RL agent needs an order of 1 x 10^8 steps to learn and generalize across tasks. However, an LLM agent can produce a control policy within 15-20 steps of learning. The point you are bringing is the exact reason why we created this benchmark.
> > >
> > > [1] Yu, Tianhe, Deirdre Quillen, Zhanpeng He, Ryan Julian, Karol Hausman, Chelsea Finn, and Sergey Levine. "Meta-world: A benchmark and evaluation for multi-task and meta reinforcement learning." In Conference on robot learning, pp. 1094-1100. PMLR, 2020.
> > >
> > > > But it is not clear how the selected environments and the constructed feedback provide these interesting properties. Would be curious if there was more careful thought in the selection of these particular environments.
> > >
> > > The guiding principle of our task selection is based on the properties of the Markov Decision Process – action space, observation space, and the interaction horizon. We want to see how feedback can impact the difficulty/easiness of learning in these varied sets of environments. Indeed studying environments that are most impacted by human feedback is an interesting direction, we are happy to investigate more in future work.
> > >
> > > > Is there an error in Appendix F.2? "Example Feedback" includes a "Thought: From the feedback, it seems that..."
> > >
> > > Thank you for pointing this out. We copied and pasted part of the LLM agent generated output into the paper draft. The implementation of the environment is correct.
> > >
> > > Hope we have answered your questions. We do believe this is a work that’s worth sharing with the larger audience, to inspire further investigations in the RL and agent community. Please let us know if you have other concerns.

---

### Official Review · Reviewer_v9Mt · 2024-11-02

**Soundness:** 3
**Presentation:** 3
**Contribution:** 2
**Rating:** 6
**Confidence:** 3

**Summary:**

This paper introduces a new benchmark designed to provide language model feedback across 8 diverse decision-making tasks. The benchmark is implemented on the OpenAI Gym framework, by modifying observation space with adding task-specific instructions and feedback text. Language feedback is generated based on reward, anticipated future feedback, and hindsight feedback, structured through task-specific templates.

**Strengths:**

Recognizing the potential of human feedback to guide agents in leveraging real-world data more effectively, this benchmark explores a valuable direction in agent learning. The paper presents extensive experimental results, utilizing both open-source and proprietary language models, highlighting the challenges for many current LLMs. The authors provide detailed descriptions of each task, including observation and action spaces, which enhances the clarity of the work. The selected tasks cover a wide range of observation and action modes.

**Weaknesses:**

While learning from language feedback could be valuable for tasks requiring human interaction, the current paper relies on a limited set of 4-20 language templates per task, which may lack the diversity needed to effectively mimic authentic human feedback. Additionally, other benchmarks, such as MINT bench, already provide language feedback during interactions, which weakens the novelty of LLF-Bench. The selection of tasks may also be less aligned with current missions of agent-human interaction, further impacting the applicability of the language feedback approach.

**Questions:**

1. **Task Selection for Rich Human Feedback**: To fully utilize the potential of language feedback in human-to-human learning contexts, would it be more effective to include tasks that are inherently cooperative? For example, humans have powerful intuitions on space and physics, which encourages more visual and physics related tasks, such as the current Metaworld task.

2. **Varied Levels of Language Feedback**: Could the paper explore different levels of language feedback detail? For example, as shown in Figure 1, level-1 feedback might provide an abstract comment such as "The arm is moving too fast/wrong direction"; level-2 feedback could include directional guidance, such as "The arm should move slightly left"; level-3 feedback might provide detailed statistics, as in the example. This approach would better mimic the often abstract nature of real-world human feedback, potentially adding practical value to the benchmark.

3. **Diversity in Language Feedback**: Generating feedback from a limited set of templates lacks the diversity seen in feedback generated by large language models, which can be easily prompted to create varied responses in different styles and with minimal repetition. Could using an LLM to generate feedback improve feedback diversity and realism? For example, using LLMs to first generate more templates, directly using LLMs with different seed prompts to generate diverse feedbacks, or using different LLMs to generate feedbacks.

4. **Effect of Vision Input on Performance**: In Table 4, adding vision input at the input stage seems to harm performance. Even though the authors claim vision input would help after removing future reward, there are still three tasks with significant performance drop. Could the authors provide an explanation for why vision input seems to negatively impact outcomes in these tasks?

---

> ### Author Response · Authors · 2024-11-23
>
> Thank you for your feedback! Our answers to your questions are as follows:
>
>
> > ...aligned with current missions of agent-human interaction...
>
> These missions can be understood in at least two ways: (a) a person teaching an AI agent do a task for them and (b) a person and an AI agent working together to accomplish a task. Both interpretations are being actively pursued by the community, and ours fall under (a): LLF-Bench is meant to assess agents’ ability to learn a task from human feedback.
>
>
> > **1. Task Selection for Rich Human Feedback:**
>
> Per the above clarification, our benchmark is meant to model situations where a human is trying to automate a task by teaching its AI assistant in natural language to perform this task. So, the scenarios our benchmark models are indeed cooperative in the sense that the agent’s incentive is fully aligned with the human one.
>
> > **2. Varied Levels of Language Feedback**
>
> This is indeed one of the motivation behind this benchmark's design. The levels of feedback in LLF-Bench already work as you propose. In particular, your Level-1 feedback is what we call “hindsight negative (HN)” feedback, and Level-2 is “future positive (FP)”. In experiments (Table 1 an Table 2), we compare how agents learn with and without Future Feedback.
>
> >  **3. Diversity in Language Feedback**
>
> Good suggestion, we will provide a wrapper for generating feedback on the fly with an LLM by seeding the response with existing templates. At the same time, we will keep the current method of choosing from hand-picked templates an option, due to its higher reproducibility.
>
> >  **4. Effect of Vision Input on Performance**
>
> The vision input can indeed help an agent that is “smart” enough to use it, but our agent is very basic and has a limited 3D understanding. Please note that our aim in constructing this agent was to provide a conceptually simple baseline that can serve as a natural starting point for other researchers to improve upon.

---

> > ### Comment · Reviewer_v9Mt · 2024-11-24
> > **Official Comment by Reviewer v9Mt**
> >
> > Thank you for the response. I'll keep my score.

---

### Official Review · Reviewer_CEBc · 2024-11-04

**Soundness:** 3
**Presentation:** 3
**Contribution:** 2
**Rating:** 5
**Confidence:** 3

**Summary:**

The authors propose a new framework for evaluating LLM agents, called LLF-Bench. The framework consists of a variety of tasks that require an LLM agent to solve, and provide both scalar and textual feedback for actions taken by the agent.

**Strengths:**

The benchmark consists of diverse domains, from recommendation, to navigation, to even robotic manipulation (which to my knowledge,  is not typically used as a evaluation domain for LLM agents). The inclusion of textual feedback in addition to scalar rewards also sets it apart from existing benchmarks.

**Weaknesses:**

I think the set of tasks chosen is diverse and reasonable. However, I do think the benchmark would be much more valuable if it also included a realistic dialogue task, such as negotiation or persuasion. I think such tasks might be the primary examples where some form of RL fine-tuning is needed.

Another potentially bigger concern I have is in the empirical evaluation. While the authors compare against different LLMs, they only consider one method, namely a Reflexion-based agent. To truly understand the nuances of each considered task, I think the authors need to also evaluate different non-RL and RL based methods, including ReAct [1] and LATS [2], respectively.

[1] https://arxiv.org/pdf/2210.03629

[2] https://arxiv.org/pdf/2310.04406

**Questions:**

(1) While I see that including textual feedback provides value to training LLM agents, it would be nice to better understand its computational cost. Namely, it would be interesting to see how much slower generating trajectories when the environment must provide textual feedback beyond the scalar reward.

(2) As alluded to earlier, I think it is important to consider more baselines beyond just a Reflexion-based agent. That way, we can better understand which tasks are difficult and may require explicit planning or multi-step reasoning to solve.

---

> ### Author Response · Authors · 2024-11-23
>
> Thank you for your thoughtful feedback. We address the weaknesses and questions you raised below:
>
> > Inclusion of dialogue tasks
>
> We agree that dialogue tasks such as negotiation or persuasion are important and challenging areas for RL fine-tuning. However our focus in this work is on learning from language feedback in a "cooperative" setting, where the feedback generator assists the agent in learning. This is a specific subset of learning from text interaction, which includes other dialogue tasks. We believe that understanding this cooperative learning process is crucial before tackling adversarial or competitive dialogue tasks like negotiation. Thank you for the suggestion, we will clarify the setting and the relationship between LLF and dialogue tasks in the paper.
>
> > Computational Cost of Textual Feedback:
>
> In all our environments, feedback is procedurally generated using code that takes the environment state as input. This process does not introduce any noticeable computational overhead. In all of the experiments, the computation time is dominated by the agent's inference, with feedback generation time being negligible. We will clarify the computation cost of the textual feedback, thank you.
>
> > Comparison with Other Methods
>
> The primary goal of Section 5 is to establish a baseline of task difficulty rather than to compare different LLM agent architectures. That said, we conducted experiments with another agent architecture (detailed in Appendix E) and found the same empirical trends. Regarding LATS, its codebase has specifically engineered prompts for different applications, so its code cannot be directly applied here. One of the objectives of LLF-Bench is to encourage the development of learning agents that are not hand-engineered for specific applications like LATS code, and promote a more general approach to learning from language feedback.

---

> ### Comment · Reviewer_CEBc · 2024-11-27
> **Reviewer CEBc Response**
>
> Thank you for your rebuttal. I will maintain my score. Regarding introducing additional baselines such as LATS, the authors are correct in that they require hand-engineered prompts. But to my knowledge, so does Reflexion, which the authors do evaluate on. Is there a reason why Reflexion can be evaluated that the authors could clarify?

---

> > ### Author Response · Authors · 2024-11-27
> >
> > You're right that the code released with Reflexion is task specific. The Reflexion tested here is based on a task-agnostic prompt template that we wrote based on the design idea of the Reflexion paper. (The exact prompt sent to LLM is then formulated based on the `instruction` , `action`, `observation`, `feedback` returned by the LLF-Bench during interactions). Since a Reflexion agent is fairly simple, we can do that; on the other hand, LATS has a more complicated workflow requiring multiple LLM queries, for which  it is harder to write a task-agnostic version that can reliably generalize to all LLF tasks. Authors releasing only task-specific implementation is one key reason we propose LLF-Bench. We hope that by introducing a formal problem setup and a diverse set of testing tasks, we can motivate the field to start developing task-agnostic agentic algorithms and implementations.

---

### Official Review · Reviewer_5kZh · 2024-11-04

**Soundness:** 2
**Presentation:** 2
**Contribution:** 2
**Rating:** 3
**Confidence:** 4

**Summary:**

This paper studies learning from language (LLF) setup where AI agent learns from interactive language feedback. To facilitate research in this direction, the authors propose LLF-Bench, providing a reproducible setup for learning from language feedback. LLF-Bench includes diverse tasks ranging from poem writing to low-level robot control. Also, it features diverse feedback types and randomization of the environment to prevent overoptimization towards a specific setup.

**Strengths:**

1. The proposed benchmark, LLF-Bench, includes a diverse set of tasks ranging from text writing to robotics.
2. The benchmark is simple and easy to use using the commonly adopted OpenAI Gym-like API.

**Weaknesses:**

1. While “proposing a formal setup of Learning from Language Feedback (LLF)” is claimed to be one of the main contributions, LLF has been widely studied in prior work[1, 2] and the formulation in the paper does not seem to be novel compared to those.
2. The experiments mainly focus on quantitatively assessing existing language models within the benchmark, with limited analysis specific to LLF. For instance, how does LLF compare to using scalar feedback alone in exact performance? How does randomizing the textual description of task instructions and feedback impact the agent's performance?
3. The authors emphasize the challenges of designing rewards and the risk of unexpected outcomes from suboptimal rewards in reinforcement learning. However, similar issues could arise for LLF if the feedback provided is suboptimal or misaligned with the task instruction.
4. Despite being presented in main Table 4, it is not clear why experiments with image-based observations are meaningful for understanding LLF, and no justification is currently provided regarding this.

[1] Reflexion: Language Agents with Verbal Reinforcement Learning. Shinn et al. NeurIPS 2023.

[2] Yell At Your Robot: Improving On-the-Fly from Language Corrections. Shi et al. RSS 2024.

**Questions:**

1. How do agents using only scalar rewards compare to LLF agents? Does it show a slower learning speed?

2. In Figure 1, future feedback includes a very detailed coordinate for the next action. Does the environment assume the availability of optimal action in feedback?

3. In Sec 3.2, while it is mentioned that LLF-Bench implements the task instruction as part of the environment rather than part of the agent, it is not clear how this paradigm makes a technical difference in the environment and contributes to “the development of more general learning agents” as mentioned in Sec 3.2.

4. In Sec 2.2, while it is claimed that the reinforcement learning agent has to not only learn to solve the task but also learn to understand the task’s objective, this might not be the case for training agent that has language understanding capability (eg., LLM) using reinforcement learning with instruction, which is the prevalent type of agent in this paper.

---

> ### Author Response · Authors · 2024-11-25
>
> Response:  We thank you for your review and we address your responses in the order in which they arise.
>
> 1. **LLF vs Past work on language feedback**:
>
> You’re right that past works have studied using language feedback to instruct or teach an agent. We do not wish to claim we’re the first to study learning from language feedback. Instead our aim is to provide a **formal** abstraction of these problems, just as the Markov decision process was proposed to abstract sequential decision making. (We will better clarify this in the revision). Having a formal problem setup of learning from language feedback is a timely need, as this area gets more attention. One important novelty of the LLF setup here is the perspective that language feedback can be viewed as a generalization of reward in MDP, separate from the observation that the agent receives. (e.g., in the LLF setup, observations and actions can be non-text while feedback is natural language). This perspective highlights that both reward and language feedback are means (e.g. of a human) to teach an agent, which are not necessarily needed for a learned agent to execute tasks later on. Following this idea, we propose a **taxonomy of feedback** to an agent (e.g. reward, hindsight, future feedback). This insight connects the Education (for humans) literature to (AI agent) learning literature, and allows us to systematically construct comprehensive benchmark environments in LLF-Bench to study how agents learn from **different** language feedback, In comparison, previous works tie language feedback to a specific agent embodiment or applications and often *implicitly* assumes one kind of feedback (e.g. future feedback in [2]) without discussing effects of other feedback types.

---

> > ### Author Response · Authors · 2024-11-25
> >
> > 2. **Additional Experiments with Rewards**:  To answer your question about how agents learn with rewards only, we ran experiments using just rewards with gpt4o. Please see the results below. We put them by the results of Table 1 and 2 for comparison. We updated our evaluation for the bandit environment to differentiate different policies better. The results are shown below:
> >
> > | EnvName | Reward, Hindsight, Future | Reward, Hindsight | Reward|
> > |----------------------------|-----------------------------------------|-----------------------------------------|-----------------------------------------|
> > | *10ArmedUniform*           | **-1.90** (0.04)                        | -5.01 (0.29)                            | -13.54 (1.81)                            |
> > | *10ArmedGaussian*          | **-1.96** (0.08)                        | -5.90 (0.31)                            | -16.22 (1.82)                            |
> > | *Booth*                    | **-3.93** (0.68)                        | -14.26 (1.81)                           | -29.62 (4.80)                           |
> > | *McCormick*                | **-0.19** (0.05)                        | -0.26 (0.03)                            | -0.27 (0.11)                            |
> > | *Rosenbrock*               | **-1.19** (0.33)                        | -10.51 (3.55)                           | -9.45 (3.42)                            |
> > | *SixHumpCamel*             | **-0.23** (0.06)                        | -0.56 (0.07)                            | -0.31 (0.11)                            |
> > | *reco-movie*               | **-5.28** (1.55)                        | -6.98 (1.08)                            | -11.74 (1.77)                           |
> > | *parking*                  | -14.32 (0.39)                           | -14.32 (0.28)                           | **-13.94** (0.43)                       |
> > | *Haiku*                    | -6.59 (1.51)                            | **-3.98** (0.82)                        | -12.52 (2.08)                           |
> > | *Tanka*                    | -9.03 (1.49)                            | -20.68 (1.35)                           | **-6.55** (1.74)                        |
> > | *LineSylConstr*            | **-13.92** (1.33)                       | -23.97 (0.64)                           | -24.64 (0.83)                           |
> > | *gridworld*                | **1.00** (0.00)                         | 0.95 (0.02)                             | 0.50 (0.07)                             |
> > | *alfworld*                 | **0.80** (0.06)                         | 0.64 (0.07)                             | 0.56 (0.07)                             |
> > | *reach*                    | **1.00** (0.00)                         | 0.82 (0.04)                             | 0.96 (0.03)                             |
> > | *button-press-wall*        | **0.82** (0.05)                         | 0.56 (0.05)                             | 0.36 (0.07)                             |
> > | *bin-picking*              | **0.88** (0.05)                         | 0.00 (0.00)                             | 0.00 (0.00)                             |
> > | *pick-place*               | **0.68** (0.07)                         | 0.46 (0.05)                             | 0.20 (0.06)                             |
> > | *assembly*                 | **0.00** (0.00)                         | **0.00** (0.00)                         | **0.00** (0.00)                         |
> > | *push*                     | **0.86** (0.05)                         | 0.35 (0.05)                             | 0.14 (0.05)                             |
> > | *box-close*                | **0.88** (0.05)                         | 0.00 (0.00)                             | 0.00 (0.00)                             |
> > | *hand-insert*              | **0.16** (0.05)                         | 0.13 (0.03)                             | 0.08 (0.04)                             |
> > | *faucet-open*              | **1.00** (0.00)                         | 0.57 (0.05)                             | 0.54 (0.07)                             |
> > | *dial-turn*                | **1.00** (0.00)                         | 0.05 (0.02)                             | 0.10 (0.04)                             |
> >
> >
> > We observe that using just rewards leads to severely degraded performance. In contrast, using full language feedback leads to best performance. This is consistent with the finding in Table 1 and Table 2 that learning with partial/weaker language feedback (i.e., without using “fp” feedback) leads to a drop in performance. This shows that language feedback helps more than rewards, and richer language feedback helps even more as expected.

---

> ### Author Response · Authors · 2024-11-25
>
> 3. **Reason for randomizing prompts**
>
> It is well-known that LLM agents are sensitive to prompt changes; see e.g., this well-cited study of Sclar et al., 2024. For this reason, we randomize the agent’s textual description and feedback to prevent prompt hacking where an agent is particularly good with one particular prompt but not robust to meaning-preserving changes in the prompt.
>
> Sclar et al., 2024: _Quantifying Language Model’s Sensitivity to Spurious Features in Prompt Design or: How I learned to start worrying about prompt formatting_, at ICLR 2024
>
>
> 4. **Designing rewards vs designing feedback**
>
> The main challenge with designing rewards over designing language feedback is that humans think in language so giving language feedback is more intuitive to us than giving scalar numbers. Humans are particularly bad at thinking about the effects of maximizing “accumulated” rewards, which is the objective of e.g. MDP used in RL. Previous studies have shown that when defining a task using accumulated rewards, humans at times inadvertently create positive reward loops which lead to the agent running in circles and getting infinite rewards (e.g., throwing trash, putting it back in the bin, and throwing it again, etc.). We argue that this intuition of human language will lead to more aligned language feedback. There is another big advantage of language feedback over rewards in that it provides a rich learning signal as language can communicate many more bits of information than a scalar reward, which can enable designing faster learning agents.
>
> 5. **Non-text-based problems**
>
> Learning from Language feedback is not limited to tasks where observations and/or actions are texts. In fact, language feedback is particularly useful in many multimodal real-world applications such as robotics (like in [2]), driverless cars, and an OS agent. E.g., imagine telling the driverless car, “you made a wrong turn, you should have stopped at the cafe on the previous street”. Our image-based experiments are an example of these applications. Our experimental results show that basic agents lack the ability to effectively use images and more work is needed to solve these tasks.
>
>
> 6. **Availability of optimal action**
>
> Future positive assumes *suggestions* of things to do, which may not necessarily mean knowledge of the optimal action. In the llf-metaworld envs, the future positive feedback contains actions that a suboptimal expert policy would take (which cannot solve the task with 100% success). You can see Appendix F.1 for another example. There the future positive feedback part corresponds to “Make recommendations that are Western, like True Grit and The Good, the Bad and the Ugly. Identify movies that were released during 2000s or 80s, like Pearl Harbor and Black Hawk Down.”. which are helpful advice but not necessarily actions to follow exactly. We also highlight that LLF-Bench allows users to determine which feedback to provide the agent. E.g., users can study learning without future positive feedback (as we did in Table 2). Such a feature of controlling what information feedback can provide is missing in existing benchmarks. This is a key strength of LLF-Bench.
>
> 7. **Task as part of the environment vs agent:**
>
> A common problem with the design of LLM agents is prompt hacking where people find prompts that works really well for a given task but fail in general, and release task-specific implementation of algorithms which are non-trivial to extend to new applications. As a result, these benchmark performance does not reflect real-world performance. By having prompt be part of the environment and requiring users to design agents that can solve multiple environments, we prevent the agent’s limit to hack to it. We also randomly select prompts from a prompt set which further reduces prompt hacking. This means that to do well on various tasks in LLF-bench, an LLM agent must be general enough to work with prompts given the environment across a range of tasks.
>
> 8. **RL vs learning from instruction:**
>
> By RL here, we imply that classical definition that agents learning from observation, action, and reward -- without any language feedback or instruction. We will clarify this in the revision.

---

> > ### Comment · Reviewer_5kZh · 2024-11-25
> >
> > Thank you for your response and the additional experiment with a reward-only baseline. Still, I think the definition of LLF is neither mathematically formal nor as clear as other frameworks (e.g., Markov Decision Process), which limits the contribution. While I acknowledge the effort put into building benchmarks with a diverse set of tasks in LLF setup, incorporating more baselines (e.g., traditional reinforcement learning agents such as PPO) and conducting further experimental analyses (e.g., sample efficiency comparison between LLF and RL agent) seem necessary to strengthen the results. Therefore, I will keep my score.

---

> ### Author Response · Authors · 2024-11-25
>
> > the definition of LLF is neither mathematically formal
>
> Thanks for the response. However, we strongly disagree with this statement. The LLF setup is formally defined in the paper, in Sec 2.1 (starting at line 110). It's sequential decision problem of [instruction, observation_1, action_1, feedback_1, observation_2, action_2, feedback_2...], which is in contrast to RL of [observation_1, action_1, reward_2,, observation_2, action_2, reward_2...]. We also describe taxonomy of feedback and how this interaction structure can enforced in the gym API in Appendix C.
>
> MDP is a Markovian version of sequential decision problems. Similarly, one can say an LLF problem is Markovian when the observation represents the state and the feedback is generated purely based on the current state.
>
> >  sample efficiency comparison between LLF and RL agent
>
> We want to highlight that in LLF we're targeting at ultra efficient learning which the traditional RL setup cannot approach. Here we show agents learning to solve the problem within a **single** episode of 15-20 steps of learning. This is a regime where no RL algorithms can touch. For example, for Meta-World [1] (see their Figure 12, 13, 14), an RL agent needs an order of 1 x 10^8 steps across **multiple** episodes to learn and generalize across tasks. In particular, the PPO algorithm you mentioned only performs update after at least one episode. So if we were to run it in the current experimental setup of one-episode, no learning would happen and the evaluation score would be just of the initial random policy.

---

### Meta-Review · Area_Chair_NELb · 2024-12-18

**Metareview:**

**summary**

This paper introduces LLF-Bench, a new benchmark for evaluating AI agents' ability to learn from language feedback (LLF). LLF-Bench encompasses a diverse set of 8 tasks, including movie recommendation, poem writing, numerical optimization, and robot control. LLF-Bench replaces numerical rewards with natural language instructions and feedback, delivered through manually crafted templates that ensure diversity and prevent overfitting. Feedback types include scalar and textual responses, generated based on reward, anticipated future outcomes, and hindsight evaluations. Built on the OpenAI Gym framework, LLF-Bench provides a reproducible environment for exploring interactive language learning, supporting a wide range of applications and research in this domain.

---

**strengths**

* Interesting topic: The benchmark addresses the growing interest in Learning from Language Feedback (LLF), filling a gap in systematic evaluation environments.
* Diverse tasks and domains: LLF-Bench includes a broad range of tasks, from creative text generation (e.g., poem writing) to technical domains like robotics and numerical optimization.
* Reproducibility: The use of manually written templates for generating feedback ensures reproducibility, cost-efficiency, and consistency, avoiding potential issues with LLM-generated feedback. Also, LLF-Bench adopts a simple and familiar API, modeled after the widely-used OpenAI Gym, which makes it accessible and extensible.

---

**weaknesses**

* Similar prior work: Benchmarks like MINT Bench already provide language feedback, diminishing the novelty of LLF-Bench
* More realistic tasks like dialogue are required.
* Limited diversity in language templates (4–20 per task) undermines the authenticity of mimicking real-world feedback scenarios.
* The benchmark includes only one method (Reflexion-based agent), lacking comparison with alternative RL and non-RL methods like ReAct and LATS

---

**decision**

Even though the authors addressed some concerns regarding insufficient experiments, several reviewers are still not fully satisfied and convinced by current results. Due to this, I'd like to suggest "reject".

**Additional Comments On Reviewer Discussion:**

The authors addressed concerns about novelty and insufficient experiments by providing additional experiments (e.g., comparison with reward feedback, new baseline, and so on). However, concerns have not been fully addressed. I recommend the authors to address the concerns above and resubmit to another venue.

---

### Decision · Program_Chairs · 2025-01-22

Reject